# Extremal Dependence between Temperature and Ozone over the Continental U.S.

Pakawat Phalitnonkiat[1], Peter G. M. Hess[2], Mircea D. Grigoriu[3], Gennady Samorodnitsky[4], Wenxiu Sun[2], Ellie Beaudry[2], Simone Tilmes[5], Makato Deushi[6], Beatrice Josse[7], David Plummer[8], and Kengo Sudo[9]

[1]Center for Applied Math, Cornell University
[2]Department of Biological and Environmental Engineering, Cornell University
[3]School of Civil and Environmental Engineering, Cornell University
[4]School of Operations Research and Information Engineering, Cornell University
[5]Atmospheric Chemistry Observations & Modeling Laboratory
[6]Meteorological Research Institute (MRI), Tsukuba, Japan
[7]CNRM UMR 3589, Météo-France/CNRS, Toulouse, France
[8]Environment and Climate Change Canada, Montréal, Canada
[9]Graduate School of Environmental Studies, Nagoya University, Nagoya, Japan

*Correspondence to:* Peter G. M. Hess (pgh25@cornell.edu)

**Abstract.** The co-occurrence of heat waves and pollution events and the resulting high mortality rates emphasizes the importance of the co-occurrence of pollution and temperature extremes. Through the use of extreme value theory and other statistical methods tropospheric surface ozone and temperature extremes and their joint occurrence are analyzed over the United States during the summer months (JJA) using measurements and simulations of the present and future climate and chemistry. Five simulations from the Chemistry Climate Model Initiative (CCMI) reference experiment using specified dynamics (REFC1SD) were analyzed: the CESM1 CAM4-chem, CHASER, CMAM, MOCAGE and MRI-ESM1r1 simulations. In addition, a 25-year present-day simulation branched off the CCMI REFC2 simulation in the year 2000 and a 25-year future simulation branched off the CCMI REFC2 simulation in 2100 were analyzed using CESM1 CAM4-chem. The latter two simulations differed in their concentration of carbon dioxide (representative of the years 2000 and 2100) but were otherwise identical. In general regions with relatively high ozone extremes over the U.S. do not occur in regions of relatively high temperature extremes. A new metric, the spectral density, is developed to measure the joint extremal dependence of ozone and temperature by evaluating the spectral dependence of their extremes. While in many areas of the country ozone and temperature are highly correlated overall, the correlation is significantly reduced when examined on the higher end of the distributions. Measures of spectral density are everywhere less than about 0.35, suggesting that at most only about a third of the time do extreme temperatures coincide with extreme ozone. Two regions of the U.S. have the strongest measured extreme dependence of ozone and temperature: the Northeast and the Southeast. The simulated future increase in temperature and ozone is primarily due to a shift in their distributions, not to an increase in their extremes. The locations where the right-hand side of the temperature distribution does increase (by up to 30%) are consistent with locations where soil-moisture feedback may be expected. Future changes in the right-hand side of the ozone distribution range regionally between +20% and -10%. The location of future increases in the high-end tail of

the ozone distribution are weakly related to those of temperature with a correlation of 0.3. However, the regions where the temperature extremes increase are not located where the extremes in ozone are large, suggesting a muted ozone response.

## 1 Introduction

The European heat wave of 2003, the Russian heat wave of 2010 and the extreme pollution and mortality increase that accompanied both events underlines the danger of heat waves and the accompanying air pollution. Summertime increases in temperature are expected in the next century in all climate scenarios (Collins et al. (2013) with future heat waves expected to be more intense, more frequent and longer lasting (e.g., Meehl and Tebaldi (2004)). Here we examine the relationship between temperature extremes and ozone extremes in measurements and in current and future model simulations. An analysis of the

joint extremes in ozone and temperature together may be particularly important as their joint impact on mortality is likely to be nonlinear (Wilson et al. (2014), Dear et al. (2005), Ren et al. (2008)).

Over most of the U.S. temperature is the first meteorological covariate with ozone (Porter et al. (2015)). The relation between ozone and temperature is complex: it is determined not only by temperature dependent ozone chemistry (Pusede et al. (2015)), but by other processes that correlate with temperature: for example, through meteorological factors such as stagnation events

or cloud cover (e.g., see Jacob and Winner (2009)) or through temperature dependent emissions (e.g., Weaver et al. (2009)). The ozone-temperature relationship is often measured with a linear slope (e.g., Steiner et al. (2010)). Increases in ozone with temperature have been reported in the range from 0-6 ppbv/$^{\circ}C$ depending on details of the analysis (Brown-Steiner et al. (2015)). The mechanisms accounting for variations in the ozone-temperature slope are still uncertain but can be at least partially exchanged by the emission regime: the slope generally increases as ozone precursor emissions increase (e.g., Pusede

et al. (2015)).

When it comes to extreme values, the relationship between temperature and ozone becomes more complicated (e.g., Steiner et al. (2010), Shen et al. (2016)), such that an overall linear slope fit does not necessarily capture the relationship. The extremal dependence between ozone and temperature has been explored using various methods. Sun et al. (2017) calculated the conditional probability of a high ozone day (ozone above the 90th percentile) given a high temperature day (temperature above

the 90th percentile). They found probabilities that range from approximately 50% in the northeastern U.S. to somewhat less than 20% in the western U.S. Schnell and Prather (2017) and Zhang et al. (2017) calculate the joint probability that ozone and temperature are extreme (above the 95th percentile) compared to the probability that either one of them is high. Schnell and Prather (2017) find that high temperature and high ozone events co-occur up to 50% of the time over the Northeast U.S. between April 1 and September 30. Zhang et al. (2017) obtain a qualitatively similar geographic pattern in the joint extremes

of temperature and ozone.

Here we propose a new method to measure the joint extremes of temperature and ozone based on the spectral dependence of the extremes. Changes in the future relation between ozone extremes and temperature extremes depend on i) changes in the nature of future temperature extremes and ii) the impact of these extremes on ozone. Globally the future increase in extreme temperatures (temperatures at the 20-year return period) in CMIP3 are similar to the increase in mean temperature (Seneviratne et al. (2012)), although there are some important regional exceptions. This would suggest that for the most part the future temperature probability distribution simply shifts to higher temperatures but does not change in shape consistent with measured trends (McKinnon et al. (2016)). On the other hand, many studies suggest that the ozone distribution will increase predominantly on the high-end due to changes in climate (e.g., see Weaver et al. (2009)) although not all (e.g., Rieder et al. (2015)). Wu et al. (2008) find increases in the high end of the probability distribution of both temperature and ozone in the midwestern U.S. in 2050, attributing this to an increase in stagnation episodes with soil-moisture feedbacks impacting the temperature distribution. At high enough temperatures (>312 K), Steiner et al. (2010) find that the ozone increase with temperature is suppressed. Steiner et al. (2010) hypothesize that this is due to the diminished role of PAN chemistry and isoprene emissions at high temperatures. Shen et al. (2016) show that ozone suppression at high temperature occurs at 23% of the CASTNET sites but hypothesize that the suppression is meteorologically induced. Based on the statistical relationship between ozone and temperature in the present climate, Shen et al. (2016) predict that future temperatures will lead to an average increase of 2.6 ozone violations per year in 2060 across the U.S. Meehl et al. (2018) examine the impact of an increase in future heat waves on ozone in two sets of future simulations: one with changing anthropogenic precursor emissions following RCP6.0 and one where the anthropogenic emissions remain fixed.

This article is organized as follows: in Section 2, we describe the datasets and model simulations used; in Section 3, we introduce the statistical procedures used to quantify the relationship between ozone and temperature. In Section 4, we present the results then discuss these results in Section 5. Section 6 gives the conclusions.

## 2    Data and Model Descriptions

In this study we examine the simulated and measured relationship between ozone and temperature extremes over the U.S. In particular we analyze a number of specified dynamics REFC1SD simulations from the chemistry-climate model initiative (CCMI) (see Eyring et al. (2013b)) for the period from 1992-2010. This allows a robust evaluation of simulated ozone and temperature extremes using analyzed meteorological fields and changing emissions against measurements. We also examine the impact of climate change on ozone and temperature extremes, comparing simulations of the current and future climate with fixed emissions. These latter are free-running simulations in that the meteorology, sea surface temperatures (SST) and sea-ice are calculated internally within the simulations. The free-running simulation of the current climate is compared with the REFC1SD simulations and the available measurements.

Most of the analysis in this paper emphasizes simulations with the Community Atmospheric Model with chemistry (CAM4-chem) within the Community Earth System Model (CESM1) Lamarque et al. (2012). Brown-Steiner et al. (2015) evaluate both the specified dynamics and free-running model configurations of the CAM4-chem against measurements over the U.S.,

including comparisons of ozone return periods. The horizontal grid resolution in the CESM1 simulations analyzed here is $1.9° \times 2.5°$; the free-running simulations have 26 vertical levels while the CESM1 REFC1SD simulation has 56 vertical levels. The CESM1 REFC1SD simulation (see Tilmes et al. (2016)) uses analyzed meteorological data from Modern-Era Retrospective analysis for Research and Applications (MERRA) from 1992-2010 and time changing anthropogenic and biomass burning

emissions as specified in Table 1. In the supplement we include results from the REFC1SD simulations in an additional four models: the CHASER, CMAM, MOCAGE and MRI models. The number of REFC1SD models analyzed is limited to those with sufficient output to derive the maximum daily temperature and the maximum daily 8-hour average ozone concentrations (MDA8). In the CMAM and MRI simulations both MDA8 ozone and daily maximum temperature are available daily; in the CHASER and MOCAGE simulations only daily MDA8 ozone data is available. Details on the additional model simulations is

given in Morgenstern et al. (2017).

The analysis of how extremes change with climate is limited to the CESM1 simulations. In the present day free-running CESM1 simulation (the GCM2000 simulation) the CO2 concentration is specified at 369 ppm, representative of the year 2000; in the future simulation (the GCM2100 simulation) the CO2 concentration is specified at 669 ppm, representative of the 2100 concentration of CO2 in the representative concentration pathway 6 (RCP6) (see Table 1). The concentrations of all other

greenhouse gases including methane are fixed at their year 2000 concentrations in both these simulations. Biogenic emissions are also fixed and are representative of the year 2000. Both the GCM2000 and GCM2100 simulations are 25-year simulations branched off the CCMI CESM REFC2 simulations in the year 2000 and the year 2100, respectively (Tilmes et al. (2016)). The first 5 years of each simulation are used as spin-up with the latter 20 years analyzed. The global mean temperature change over the continental U.S. between GCM2000 and GCM2100 is $2.1°C$, while the temperature difference in the parent CCMI REFC2

simulations following RCP6 is $2.8°C$. The smaller temperature increase between the GCM2000 and GCM2100 simulations is likely due to the fact that the emissions of GHGs and short-lived forcing agents are held constant at the year 2000 levels in both simulations. In particular the aerosol emissions remain the same.

Hourly measured ozone and temperature data are taken from 23 CASTNET (Clean Air Status and Trends Network) stations with a nearly continuous data record during the period from 1992-2013 for the months of June, July, and August (92 days

each summer). In addition, to enhance the data record, we included two additional stations (Beufort NC, and Lassen Volcanic CA) where the first 2 years or 3 years of data were missing, respectively. See Figure 2 for station locations. CASTNET sites are situated to sample regional ozone concentrations so as to minimize the more local impact of urban areas. We supplement the CASTNET data with temperature and ozone measurements from the Environmental Protection Agency (EPA) Air Quality System (AQS) Data Mart for the years 1992-2010. This gives an additional 124 stations with nearly complete ozone and

temperature data (see supplement). Ozone data from the first model level provides a good estimate of 10-meter ozone concentrations as measured by CASTNET (Brown-Steiner et al. (2015)). The maximum daily 2-meter temperature is used in both the CASTNET measurements and the simulations.

To render the data approximately stationary on both the interannual and seasonal basis, we adopt the procedures in Phalitnonkiat et al. (2016). Formally, let $x_{y,d}$ represent the data on day $d$ in year $y$, where $x$ refers to either daily maximum

temperature or MDA8 ozone. Since there are 91 days included in each summer period and 20 years (19 years for REFC1SD), $d = 1$ refers to June $1^{st}$ and $d = 91$ refers to August $30^{th}$.

To minimize year-to-year variability so as to minimize any ozone trends while still keeping extreme data relevant, for each year $y$, we take the average of the data over that year but omit a number ($a$) of the highest values. That is, for a fixed year $y$, the resulting average is $m_{y,a}$:

$$m_{y,a} := \frac{1}{D-a} \sum_{i=1}^{D-a} x_{y,(i)} \tag{1}$$

where $D = 91$ is the total number of days for each year, and $x_{y,(i)}$ is the order statistic of the fixed year $y$: $x_{y,(1)} \leq x_{y,(2)} \leq ... \leq x_{y,(D)}$. Then, we calculate a daily ozone deviation:

$$\hat{x}_{y,d}^G = x_{y,d} - m_{y,a}. \tag{2}$$

In our analysis, we use $a = 10$ as the default value which preserves about 11% of the extreme data. Sensitivity tests at a number of stations suggest the result is not sensitive to $a$. To eliminate seasonal effects, we average $\hat{x}_{y,d}^G$ for each day $d$ over all years $Y = 20$ (or $Y = 19$ for REFC1SD). That is, for each day $d$, we calculate:

$$M_d = \frac{1}{Y} \sum_{y=1}^{Y} \hat{x}_{y,d}^G. \tag{3}$$

$M_d$ is then smoothed by local polynomial regression since our sample size is rather small. In order not to overburden the notation, we will still use the notation $M_d$ for the smoothed values of the estimates. Then we normalize the data by

$$\hat{x}_{y,d}^{DS} = \frac{\hat{x}_{y,d}^G - M_d}{sd_d}, \tag{4}$$

where $sd_d = \sqrt{\frac{1}{Y} \sum_{y=1}^{Y} (\hat{x}_{y,d}^G - M_d)^2}$ is the standard deviation of day $d$. Later in the text, we refer to (4) as a normalized scale.

In addition to the transformations from Phalitnonkiat et al. (2016), we add another procedure to revert the normalized scale data back to its original scale while keeping the stationarity. That is, we rescale $\hat{x}_{y,d}^{DS}$ back to its original scale ($\hat{x}_{y,d}^{res}$) by using the formula:

$$\hat{x}_{y,d}^{res} = \hat{x}_{y,d}^{DS} \times \left( \frac{1}{D} \sum_{d'=1}^{D} sd_{d'} \right) + \frac{1}{D} \sum_{d'=1}^{D} M_{d'} + \frac{1}{Y} \sum_{y'=1}^{Y} m_{y',a}, \tag{5}$$

where $Y = 20$ (or $Y = 19$ for REFC1SD).

## 3  Methodology

In this study besides using conventional methods, such as correlations, to quantify the relationship between temperature and ozone we also propose a novel metric to capture the relationship between ozone and temperature extremes. Correlation coefficients are inadequate for capturing the relationship between the extremes of two variables since they are estimated from all

observations and extremes represent a small percentage of these observations. An alternative metric is proposed using only the largest values of two variables. After some transformations which act to normalize the two variables (see B), their extreme dependency is characterized by a probability density function (pdf) that measures the angular density when the variables are plotted against each other. The area under the pdf is 1 by definition and the range of the pdf is $\left[0, \frac{\pi}{2}\right]$. If the mass of the pdf is concentrated near 0 or near $\frac{\pi}{2}$, extremes of the two variables are unlikely to be significant at the same time, which points to an independence of the extremes. On the other hand, if the mass of the pdf is concentrated away from the endpoints $0, \frac{\pi}{2}$, then simultaneous extremes of the two variables are likely. We refer to the procedure which normalizes the tails of the data so that the method described above works *the ranks method* (see B).

Since the area under the curve from 0 to $\pi/2$ is 1, we can consider only the area of the 'middle' part, which we define to be between $\frac{\pi}{8}$ and $\frac{3\pi}{8}$, to represent the extreme dependence between two variables. Denote this amount by $\varphi$:

$$\varphi := \text{area} \left[\frac{\pi}{8}, \frac{3\pi}{8}\right]. \tag{6}$$

See the detailed explanation in Appendix C. Note that the range of $\varphi$ is $[0, 1]$, where $\varphi = 1$ refers to extreme dependence and extreme independence implies $\varphi = 0$.

Figure 1 shows different scenarios of correlation and extreme dependence. Figure 1a gives a scenario of data with high correlation, yet the extremes of the data are only moderately dependent (Figure 1c). In contrast, Figure 1b gives an example of data with low correlation but highly dependent extremes (Figure 1d).

## 4    Results

In this section we compare measured and simulated temperature and ozone records separately (4.1) and then their joint dependence are analyzed in section (4.2). The extremes of ozone and temperature and their extremal dependence is emphasized. Simulated ozone and temperature records from the REFC1SD, GCM2000 and GCM2100 CESM1 simulations are given in the main body of the paper. Simulated results from the REFC1SD simulations for the CHASER, CMAM, MOCAGE and MRIs models are given in the supplement. In the main body of the paper all the measurements shown are from CASTNET. Addition measurements at the AQS sites are given in the supplement. For any given simulation, all percentiles are given with respect to that particular simulation. In particular, percentiles for the future simulations are given in terms of the future distributions. Note, in addition, that all quantities shown have been rescaled following equation (5).

### 4.1    Separate Evaluation of Temperature and Ozone

The highest rescaled average daily maximum temperatures naturally occur in the South with local maximum in the Southwestern U.S., the Midwestern region and the East coast (Figures 2a, c, e; Figure S1). The simulations do not represent the topography with the accuracy adequate to simulate temperatures in regions of large topographic relief characteristic of the Western U.S. Overall, when evaluated at the CASTNET sites, temperature is slightly underestimated in the REFC1SD simulations and slightly overestimated in the CESM1 GCM2000 simulations (see Table 2). The CMAM and MRI REFC1SD

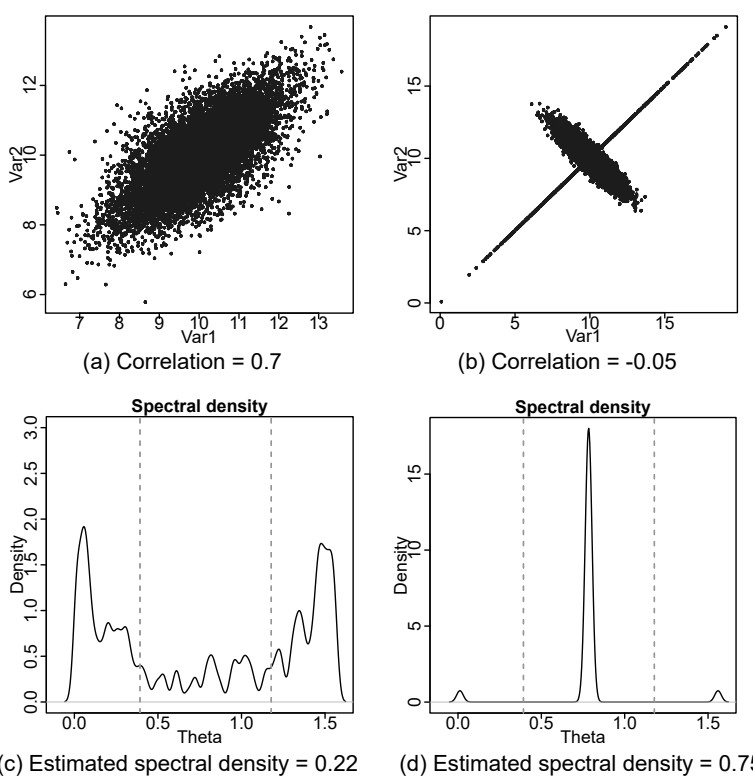

**Figure 1.** Examples that shows the correlation and extremal measure of dependence between two variables are not necessarily the same. The plots on the left column (a,c) use the data generated by Gaussian random vectors with correlation $\rho = 0.7$ and each component has $n = 10000$ points sampled from $N(10, 1)$, where $N(\mu, \sigma^2)$ is the normal distribution with mean $\mu$ and variance $\sigma^2$. The data are moderately correlated, while it has low extreme dependence (true $\varphi = 0$; estimated $\varphi = 0.223$). The plots on the right column (b,d) use the data generated by $(Var_1, Var_2) = (Y_1, Y_2)$ with probability 0.8 and $(Var_1, Var_2) = (Z, Z)$ with probability 0.2, where $(Y_1, Y_2) \sim N\left(\mu = [10, 10]^T, \Sigma = [1, -0.9; -0.9, 1]\right)$ follows a bivariate normal, where $\mu$ is the mean vector and $\Sigma$ is the covariance matrix, and $Z \sim N(\mu = 10, \sigma^2 = 9)$. The sample size is also $n = 10000$. The plots show the existence of tail dependence by having high angular density near $\frac{\pi}{4}$ ($\varphi = 0.75$); however, low correlation (true $\rho = 0$; estimated $\rho = -0.05$).

simulations have large positive biases in mean temperature (Figure S1). The spatial correlation between measured and simulated rescaled temperature in the CESM1 REFC1SD and GCM2000 simulation is between 0.57 and 0.53 respectively. In the GCM2100 simulation, rescaled maximum daily temperature increases by $2.43°C$ on average over the U.S. compared with the GCM2000 simulation (Figures 2c, e and Table 2), where the regions of high temperatures in the GCM2100 simulation expand prominently with a tongue of the highest temperatures extending throughout the Midwest.

In all simulations the width of the high end maximum daily temperature distribution, calculated as the difference between the $90^{th}$ percentile and the average maximum daily temperatures (i.e., $mean(T|T > 90\%) - mean(T)$) maximizes in the northern part of the domain, but with a tongue of high temperatures differences extending southwards through the Midwest (Figure 2b, d, f; Figures S1b, d). Both the REFC1SD and GCM2000 simulations underestimate $mean(T|T > 90\%) - mean(T)$, whereas the CMAM and MRI simulations are relatively unbiased (Figures S1b, d). There is some evidence of a similar pattern to that simulated in the CASTNET measurements and AQS measurements. Overall, the conditional maximum temperature differences show little response to climate change (e.g., compare Figures 2d and 2f) suggesting the high end of the future temperature distribution does not change markedly with respect to the mean. This is consistent with the historical changes in the temperature distributions (McKinnon et al. (2016)).

In all simulations rescaled MDA8 ozone is highest in the Southwestern U.S. and in the Middle Atlantic regions extending towards the central Midwest (Figures 3a, c, e; Figures S2a, c; Figures S3a, c). The westward extent of this ozone maximum is not reflected in the CASTNET data. Consistent with many GCMs (e.g., Lamarque et al. (2012); Rieder et al. (2015)), all the simulations have high ozone biases (Figures 3a, c; Figures S2a, c; Figures S3a, c; Table 2). Averaged over all CASTNET stations simulated surface ozone is biased high by approximately 12 ppb in the CESM1 REFC1SD simulations and 21 ppb in the GCM2000 simulation. The spatial correlation between measured and simulated ozone in the CESM1 REFC1SD simulation and the GCM2000 simulation is 0.24 and 0.23 respectively (with $p$-value at 0.25 and 0.26, respectively for the alternative hypothesis of the correlation not being 0). In the GCM2100 simulation, ozone increases by approximately 2 ppb averaged over the U.S. with respect to the GCM2000 simulation (Figure 3e and Table 2).

Despite the simulated positive bias in average ozone, the simulated difference between the $90^{th}$ percentile and average MDA8 ozone is biased low in the CESM1 simulations (Figures 3b, d) with average biases of -0.79 and -4.28 ppb in the CESM1 REFC1SD and GCM2000 simulations, respectively. Thus the CESM1 simulations underestimate the width of the high end of the ozone distribution. Of the other REFC1SD simulations examined, only the MOCAGE simulation shows a high bias in the width of the high end MDA8 ozone distribution (Figure 2; Figure S3). In all simulations except CMAM in the southeast U.S. the overall simulated pattern is similar to the CASTNET measurements with the largest differences in the Eastern part of the domain. The geographic pattern for the high-end width of the MDA8 ozone distribution $(mean(O3|O3 > 90\%) - mean(O3))$ is significantly different from the equivalent quantity for temperature. While the width of the maximum temperature distribution $(mean(T|T > 90\%) - mean(T))$ maximizes in the central U.S. (Figures 2b, d, f; Figure S1) the width of the MDA8 ozone distribution maximizes in the Eastern U.S. (Figures 3b, d, f; Figures S2, S3). On average the difference between $90^{th}$ percentile MDA8 ozone and average MDA8 ozone in the CESM1 simulations increases only by 0.26 ppb in the future simulation (Figure 3f).

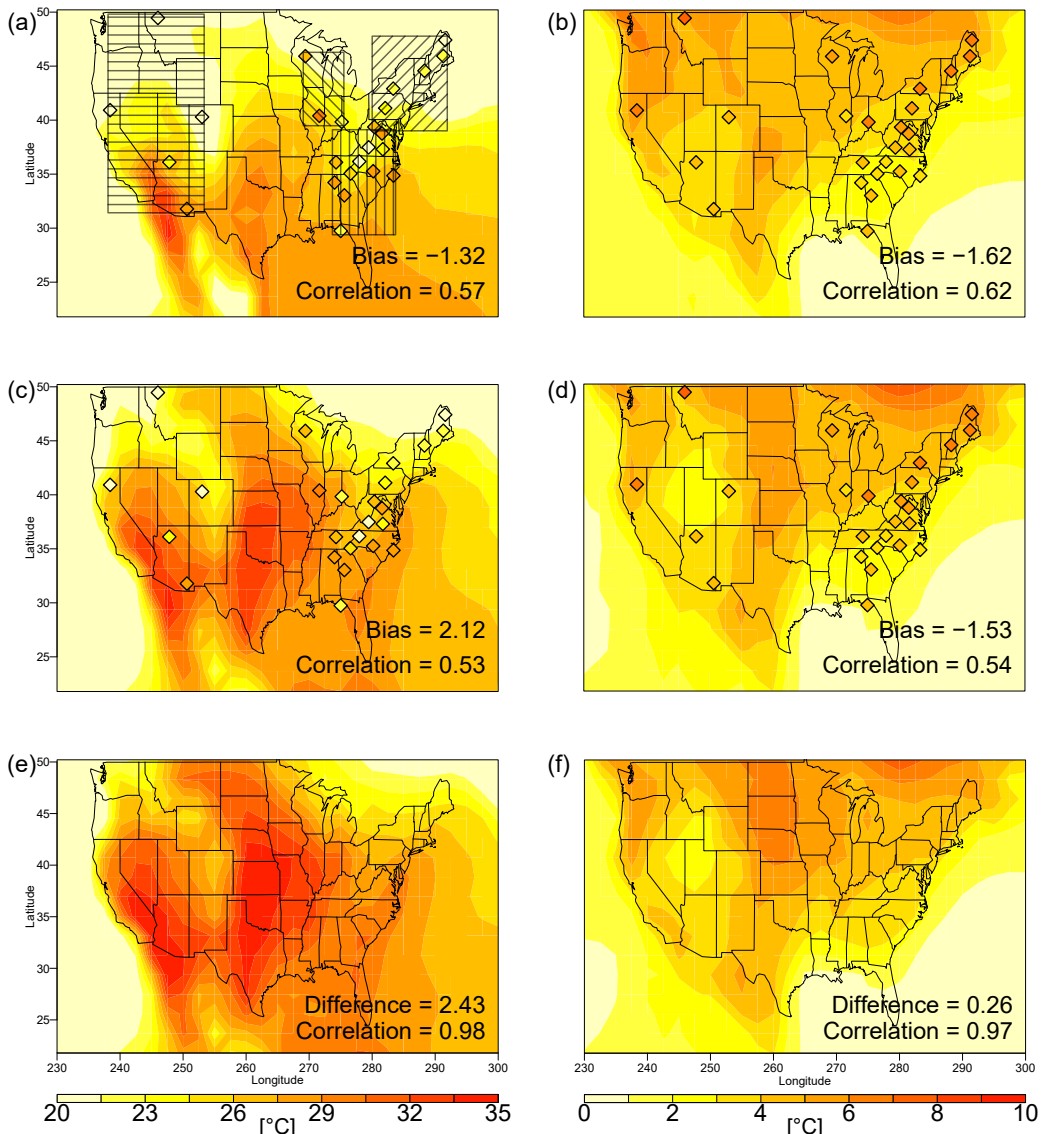

**Figure 2.** [Rescaled Data] Average daily maximum temperature (°C) (left column), and average daily maximum temperature (°C) conditioned on maximum temperature greater than the 90th percentile minus average daily maximum temperature (right column) for the CESM1 REFC1SD simulation (1992-2010) (first row), the GCM2000 simulation (2006-2025) (second row) and the GCM2100 simulation (2106-2125) (third row). CASTNET measurements (1992-2011) of each quantity are shown as filled diamonds in the first two rows. In the first two rows we also give: the average bias as the model average minus the CASTNET average for each quantity, and the correlation as the spatial correlation between the model and the CASTNET measurements. In the last row we give: the difference as the mean difference between GCM2100 and GCM2000 over the continental area between $21°N$-$51°N$ and $230°E$-$300°E$ and the correlation as the correlation between GCM2100 and GCM2000 over the continental area between $21°N$-$51°N$ and $230°E$-$300°E$. The boxes in (a) show the division of the country into various regions: the Northeast, the Southeast, the midwest and the west.

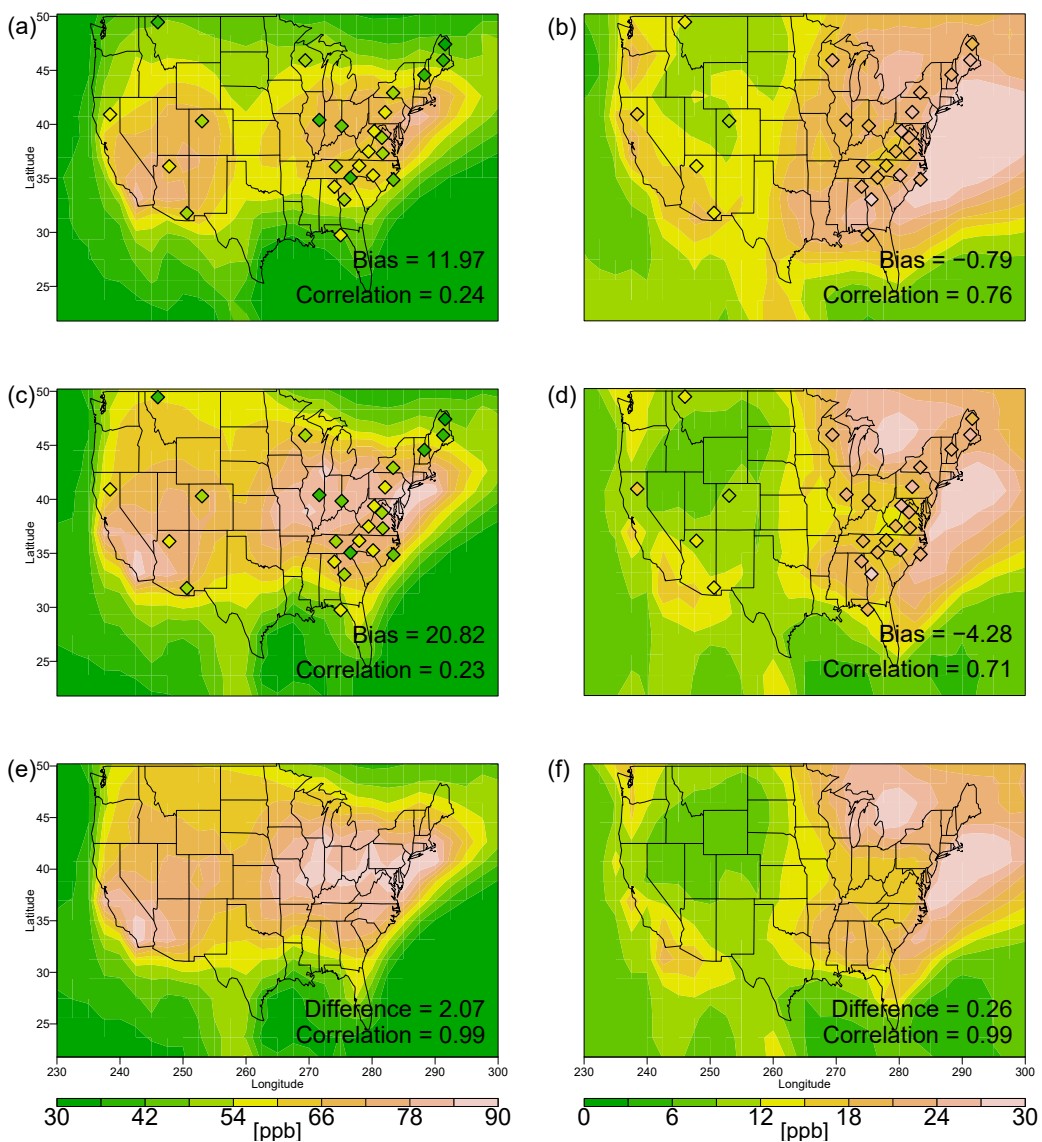

**Figure 3.** [Rescaled Data] Average MDA8 ozone (ppb) (left column), and average MDA8 ozone (ppb) conditioned on MDA8 ozone greater than the $90^{th}$ percentile minus average MDA8 ozone (right column) for the CESM1 REFC1SD simulation (1992-2010) ($1^{st}$ row), the GCM2000 simulation (2006-2025) ($2^{nd}$ row) and the GCM2100 simulation (2106-2125) ($3^{rd}$ row). The biases, differences and correlations are defined similarly to Figure 2.

The relative difference between changes in extreme values and the change in median values in the future simulation compared to the present day simulation can be expressed as the quantity $\Psi$.

$$\Psi(X,Y) := \frac{\text{Mean\_GCM2100}(X|Y > 90\%) - \text{Mean\_GCM2100}(X|45\% < Y < 55\%)}{\text{Mean\_GCM2000}(X|Y > 90\%) - \text{Mean\_GCM2000}(X|45\% < Y < 55\%)}, \tag{7}$$

where $X, Y$ are ozone or temperature. If the change in extreme increments in $X$ given $Y$ in the GCM2100 and GCM2000 simulations are the same, we expect the ratio to be 1.

The high-end width of the future maximum daily temperature distribution is projected to increase relative to the present day temperature distribution by up to 30% in the Southeast U.S. extending northwards through the Eastern Mid-west (see Figure 4b). In contrast, $\Psi(O_3, O_3)$ is less than 1 over much of the domain (Figure 4a). Note, however, that the region where $\Psi(O_3, O_3)$ is slightly greater than 1, extending from the southeast U.S. northwestward to the Midwest corresponds quite close to where $\Psi(T,T)$ is greater than 1. The overall correlation between these quantities is 0.3, significant, but weak. There have

been varying predictions for whether future ozone increases in the extreme (e.g., Sun et al. (2017)). Figure 4a suggests that in only a few locations in a future climate does the $90^{th}$ percentile ozone concentration increase by at least 10% over the increase in the median.

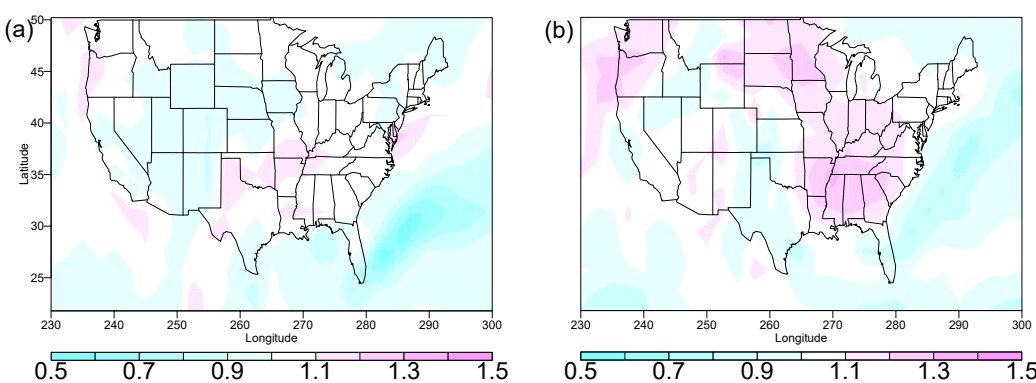

**Figure 4.** [Rescaled Data] a) $\Psi(O_3, O_3)$, b) $\Psi(T,T)$. (See the definition of $\Psi$ in (7)).

     As an alternative way of viewing the data we also present the 20-year return levels to describe the marginal extremes (Figure 5). For a stationary independent series the return level is simply related to the value at a given percentile of the distribution.

We note, however, that the 20-year return level represents a value considerably further out on the high-end of the distribution than the 90th percentile (compare Figures 2b, d, f with Figures 5b, d, f for temperature and Figures 3b, d, f and Figures 5a, c, e for ozone). Return levels are calculated using the procedure given in Phalitnonkiat et al. (2016) (see Appendix A for more detail). Differences between the 20-year return ozone MDA8 concentration and the mean concentration are generally higher in the eastern U.S. than in the western U.S. (Figure 5), consistent with Figures 3b, d, and f. This difference is underestimated

in both the CESM1 REFC1SD and GCM2000 simulations, although more dramatically so in GCM2000 and particularly in the mid-Atlantic region (Table 3). Overall, the difference between the simulated twenty-year return period ozone and the mean

ozone are biased low by approximately 3.3 ppb in the CESM1 REFC1SD simulation and 9.4 ppb in the GCM2000 simulation (Figure 5 and Table 3). This underestimation suggests that the simulations do not capture the width of the high-end MDA8 ozone distribution as measured by the 20-year return period minus the mean. Note that while the simulations underestimate the differences between the 20-year return MDA8 ozone concentration and the mean concentration the 20-year return levels are biased high (Table 3). Of the four other REFC1SD simulations examined (the CHASER, CMAM, MOCAGE, MRI simulations) all except the MOCAGE simulations underestimate the high-end tail of the ozone distribution as measured by the 20-year return period. All simulations except the MOCAGE show a relative minimum in width of the tail in the mid-Atlantic region, a minimum not captured in the measurements.

Future changes in the twenty-year return period MDA8 ozone concentration and the difference between the twenty-year return MDA8 concentration and the mean concentration between the GCM2100 and GCM2000 simulations are: 2.6 and -0.28 ppb respectively, as measured at the CASTNET sites. This result is consistent with Rieder et al. (2015), but is at odds with a number of studies that suggest future ozone levels will increase primarily at the high end due to the impact of climate (e.g., Wu et al. (2008)). The only region we find an increase at the high end is the Midwest (Table 3).

Simulated differences between 20-year maximum daily return temperatures and mean temperature (Figure 5; Figure S4) are largest in the northern part of the domain and extend southwards through the Midwest consistent with Figure 2 and Figure S1. The GCM2000 simulation generally captures the measured twenty-year return maximum temperature level while the CESM1 REFC1SD is biased low by almost $3.5^o$ C (Figure 5, Table 3). The CMAM simulation captures the width of the temperature distribution as measured by the 20-year return period of temperature while the MRI simulation is biased low (Figure S4).

The GCM2100 maximum daily temperatures (Figure 2) and 20-year maximum daily temperature return levels increase (Table 3), with a relatively small increase in the temperature difference between the twenty-year return value and the mean temperature. In the Midwest this difference increases by about $3.5^o$.

## 4.2 Joint Dependence of Temperature and Ozone

In this section, we examine the joint dependence of ozone and temperature in the simulations and in the data. In particular, we are interested in how high ozone events are related to high temperature events in the present and future climates. We use three measures to quantify this dependence and to compare it between the future and present climates and with the measurements: the ozone temperature correlation and conditional correlation, and the metrics: $\Psi$ and $\varphi$.

- We analyze the correlation between ozone and temperature to measure the overall linear correlation between these fields. We also analyze the correlation between MDA8 ozone and maximum daily temperature conditioned on maximum daily temperature greater than the $90^{th}$ percentile to measure the relationship of ozone and temperature at higher temperatures.

- The quantity $\Psi(O_3, T)$ measures the relative response (against the mean response) of MDA8 ozone at the $90^{th}$ percentile level to daily maximum temperature at the $90^{th}$ percentile level in the future versus present climate (Equation (7)).

- The quantity $\varphi$ gives an explicit relationship between ozone and temperature extremes (Equation (6)).

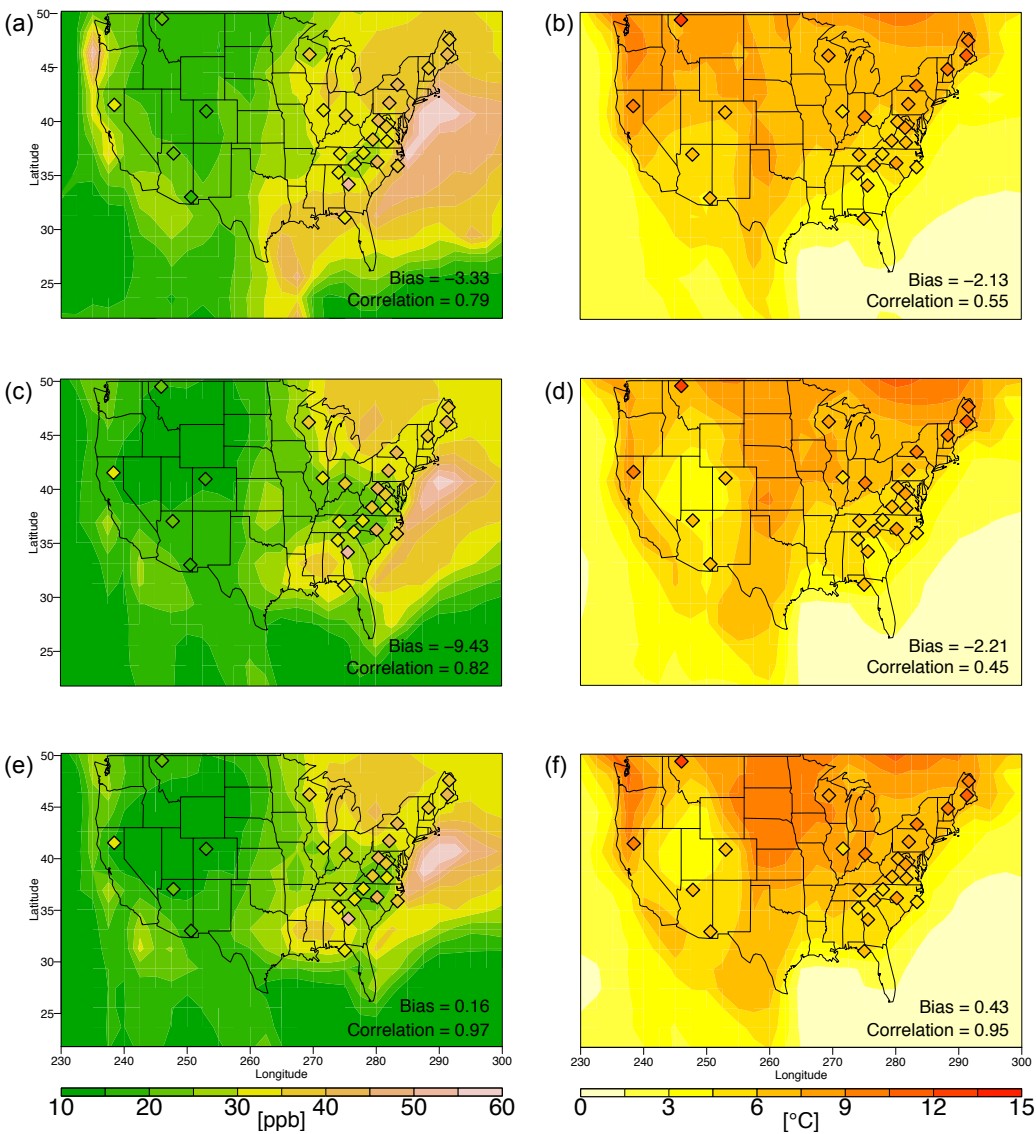

**Figure 5.** [Rescaled Data] 20-year return level minus average MDA8 ozone (ppb) (left column), 20-year return level minus average daily maximum temperature ($^{\circ}$) (right column) for the CESM1 REFC1SD simulation (1992-2010) ($1^{st}$ row), the GCM2000 simulation (2006-2025) ($2^{nd}$ row) and the GCM2100 simulation (2106-2125) ($3^{rd}$ row). 20-year return levels from CASTNET measurements (1992-2011) for each quantity are shown as filled diamonds in the first two rows. The bias and correlation in the first two rows and the difference and correlation in the last row are defined as in Figure 2.

The various simulatations between the MDA8 temperature and maximum daily ozone correlation show some similarities, but also distinct geographical differences (Figures 6a, c, f; Figure S6). All simulations have a region of low correlation within the middle of the country. The three REFCS1D simulations with high frequency temperature and ozone output (the CESM1, CMAM and MRI simulations) show this region extending inland from the Gulf of Mexico, although the CESM1 REFC1SD simulation displaces this region of low correlations further to the east than the other two. The GCM2000 and the GCM2100 simulations displace the region of low correlations further to the north without the obvious connection to the Gulf of Mexico. Differences in the correlations between the simulations are also apparent in the western and eastern thirds of the country. The MRI and CMAM simulations strong positive correlations in the northeastern U.S. extending westward and southward, while the CESM1 simulations have weaker correlations throughout the East. All three simulations using the CESM1 have a correlation maximum over the southeastern states, with the GCM2000 and GCM2100 simulations showing a relative minimum over the northeastern states. In contrast, the CMAM and MRI simulations (Figure S6) show a maximum correlation over the northeastern states extending to the northwest. All simulations show a band of high correlations over the western states, with all simulations but the CMAM simulation showing regionally high correlations over the Rockies. Based on the rather sparse CASTNET and AQS measurements, it is difficult to determine which simulation better captures the true correlation pattern.

The conditional correlations between MDA8 ozone and maximum daily temperature when maximum daily temperature is greater than the $90^{th}$ percentile are significantly reduced across the country in comparison with the unconditional correlations. Measured conditional correlations are, in all cases, marginally positive or negative. The simulated conditional correlations in the CESM1 are somewhat higher than measured, with a maximum in the Gulf coast states. The conditional correlations in the CMAM and MRI simulations are distinctly lower than in the CESM1 (Figures 6a, c, f; Figure S6).Shen et al. (2016) shows a suppression of ozone at high temperatures at many sites across the U.S.

A metric for the response of MDA8 ozone to high temperatures can be defined as MDA8 ozone (ppb) conditioned on daily maximum temperature greater than the 90th percentile minus average MDA8 ozone (Figures 7a, b, c). While the geographic extent of the measurements is somewhat limited, the measured response to this metric appears to be high in an arc extending from the northeast U.S. along the eastern seaboard into the southeastern U.S. The southeastern U.S. is also a region where the right hand side of the temperature distribution is rather narrow (see Figures 2b, 5b; Figures S1c, S4c). This suggest that ozone in the southeast U.S. is particularly sensitive to comparatively small changes in temperature. Note, however, the results do seem somewhat at odds with Shen et al. (2016) who find that temperature in the Southeast does not improve their statistical model of ozone exceedances.

The MRI simulation (see Figure S7b) captures this measured pattern the best of all the model simulations with a model-measurement correlation coefficient of 0.83. The CESM1 simulations miss the high response over the northeast U.S.: the largest simulated response in the CESM1 simulations extends off the Eastern seaboard into the southeastern U.S. (Figures 7a, b, c), but does not extend to the northeastern U.S. itself. In contrast, the CMAM simulation (see Figure S7a) shows a high sensitivity of ozone to temperature extremes over the northeast U.S., but misses the extension of the response along the eastern seaboard of the U.S. Note that the CESM1 REFC1SD and the GCM2000 simulations show similar responses in the Southeast

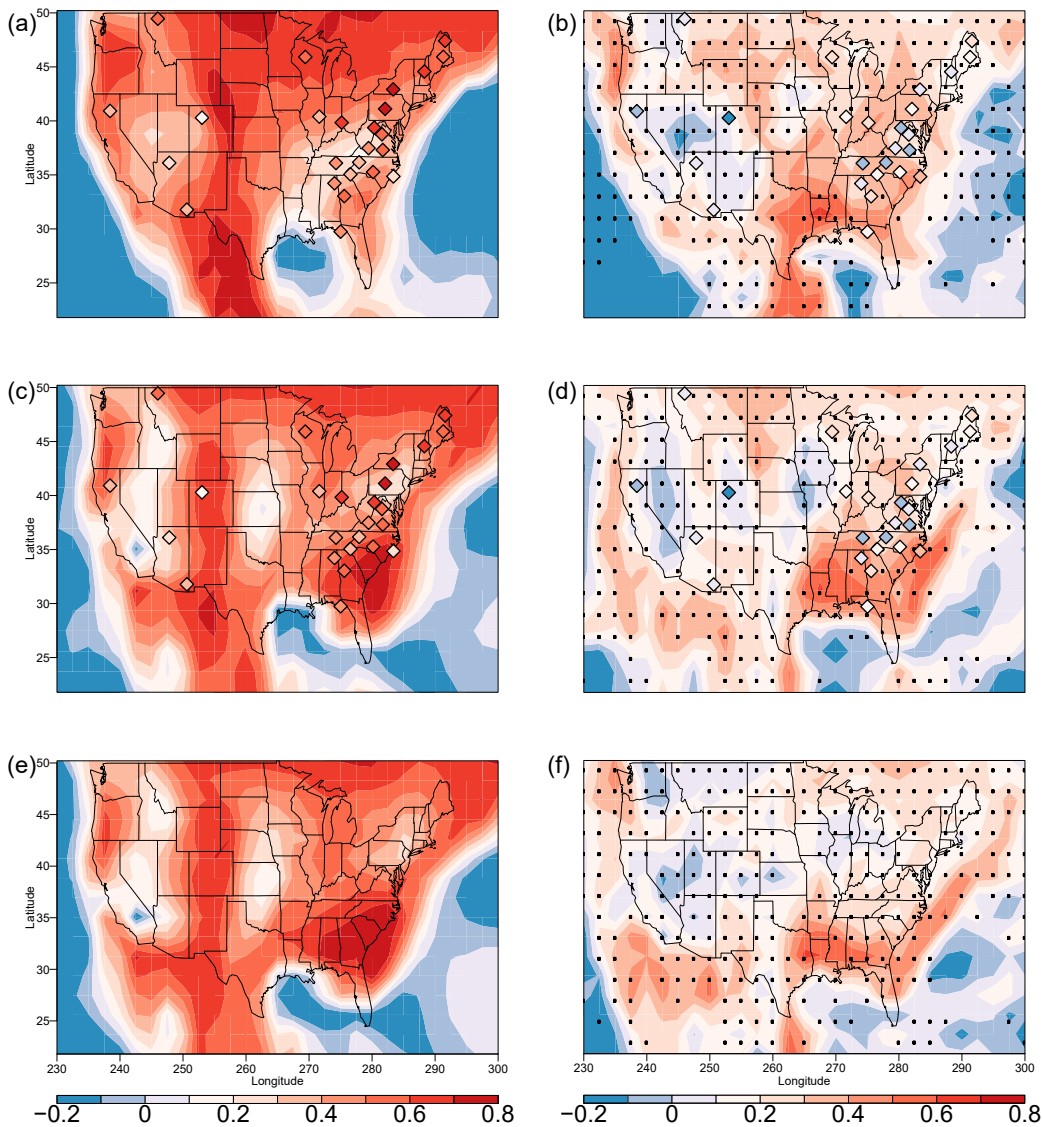

**Figure 6.** [Deseasonalized Data] Unconditional correlations between maximum daily temperature and MDA8 ozone ($1^{st}$ column); correlations between maximum daily temperature and MDA8 ozone conditional on maximum daily temperature greater than the $90^{th}$ percentile ($2^{nd}$ column), for the CESM1 REFC1SD simulation (1992-2010) ($1^{st}$ row), the GCM2000 simulation (2006-2025) ($2^{nd}$ row) and the GCM2100 simulation (2106-2125) ($3^{rd}$ row). The unconditional and conditional correlations from CASTNET measurements (1992-2011) are shown as filled diamonds in the first two rows. The black dots on the right panels indicate the significant changes from the unconditional to conditional correlations.

even though the CESM1 REFC1SD simulation includes interactive isoprene emissions, while the GCM2000 simulations does not.

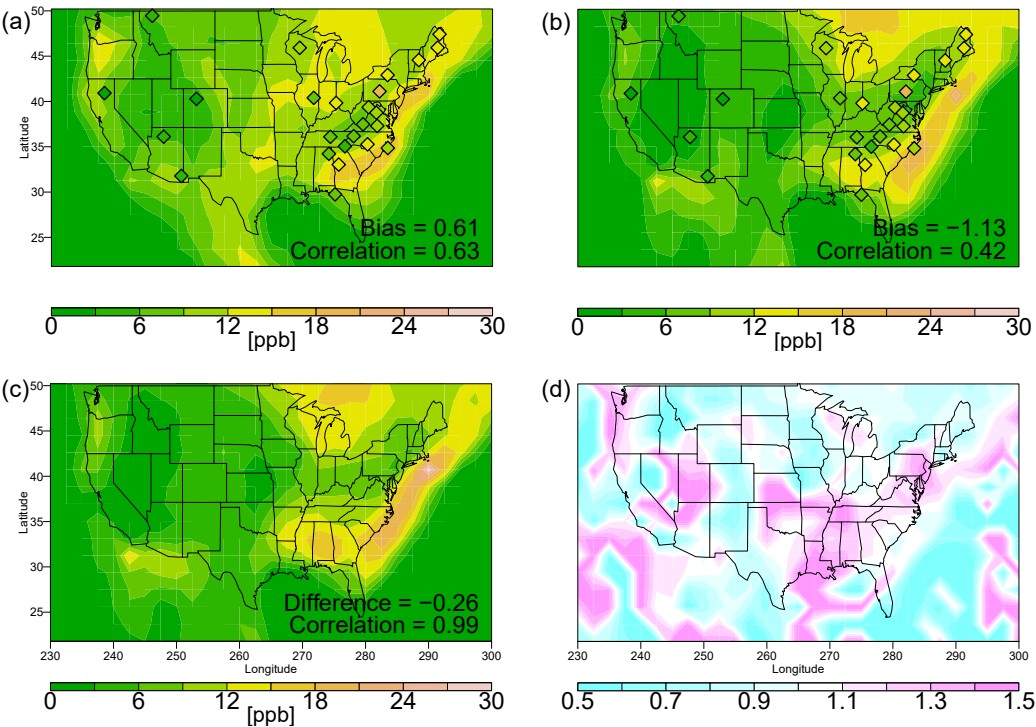

**Figure 7.** [Rescaled Data] Average MDA8 ozone (ppb) conditioned on daily maximum temperature greater than the $90^{th}$ percentile minus average MDA8 ozone for (a) the CESM1 REFC1SD simulation (1992-2010), (b) the GCM2000 simulation (2006-2025) and (c) the GCM2100 simulation (2106-2125). The biases, differences and correlations are defined similarly to Figure 2. (d) $\Psi(O_3, T)$ (See the definition of $\Psi$ in (7)).

Averaged over the continental U.S., MDA8 ozone conditioned on maximum daily temperature greater than the $90^{th}$ percentile minus average MDA8 ozone (Figure 7c) decreases modestly by 0.26 ppb between GCM2000 and GCM2100. Given
5  the 2.07 ppb future increase in mean ozone [see Figure 3] this implies ozone conditioned on the 90th percentile of mean daily maximum temperature increases by 1.81 ppb. This is consistent with a suppression of ozone at high temperatures at many sites across the U.S. The comparative sensitivity of ozone to temperature increases in GCM2100 versus GCM2000 can be assessed with $\Psi(O_3, T)$ (equation 7). While most of the country shows future decreases in temperature sensitivity a number of regions, including the gulf coast states and the Pacific Northwest, show an increase in sensitivity by over 30% (Figure 7d). Both of these
10  regions also show an increase in $\Psi(T, T)$ (Figure 4b).

Measured and simulated scatter plots of deseasonalized MDA8 ozone versus maximum daily temperature are shown for three CASTNET sites in Figure 8. In each plot, the extreme points after the normalization by the ranks method (see B for the

detailed procedure) are shown in red. As described above (Section 3) the extremal dependence between the two variables is characterized by $\varphi$, where $\varphi$ gives the proportion of the extreme points where both variables are simultaneously extreme. These sites are selected to show a range of behavior in measured and simulated $\varphi$: at one site measured $\varphi$ is larger than simulated (Ashland, Me), at one site it is less than that simulated (Sand Mountain, Al), and at one site the measured and simulated values are about the same (Beaufort, NC). At Ashland Maine (Figure 8a), the CESM1 REFC1SD, the GCM2000 and the GCM2100 simulations underestimate the extreme dependence of ozone on temperature, where about 25% of the measured points have simultaneous ozone and temperature extremes; at Sand Mountain Alabama (Figure 8b), the model simulations overestimate the extreme dependence, where approximately 9% of the measured data have simultaneous ozone and temperature extremes; at Beaufort North Carolina (Figure 8c), about 20% of the simulated and measured extremes occur simultaneously for temperature and ozone.

The measured sites where ozone and temperature extremes tend to co-occur (Figure 9) in the northeastern U.S. and in the southeastern U.S. are related to those sites where ozone shows the most response to high temperatures (Figure 7a, b). Previous studies have used different methodologies to capture the extremal dependence in measurements between ozone and temperature (Sun et al. (2017), Schnell and Prather (2017), Zhang et al. (2017)). Sun et al. (2017) finds that the conditional probability of a high ozone day given a high temperature day is approximately 50% in the northeast U.S. and 30% in the southeast and mid-Atlantic regions. Schnell and Prather (2017) also find that the co-occurrence of temperature and ozone extremes maximize over the northeast U.S. (occurring 50% or more of the time in their analysis) but decrease towards the Midwest (where joint occurrences occur 25% or less of the time). They also find a secondary maximum of less amplitude in the joint occurrence of extremes over the southeastern U.S. consistent with our analysis. However, Schnell and Prather (2017) do not find the spine of low co-occurrences clearly seen in the CASTNET data from northern Alabama to Pennsylvania (also see Figure S7f for the AQS measured data).Zhang et al. (2017). Zhang et al. (2017) find the co-occurrence of extreme ozone and temperature occurs 32% of the time averaged over the U.S. with a maximum over the northeastern U.S. during JJA and indications of a possible secondary maximum over the southeastern U.S.

None of the simulations using the CESM1 capture the measured high co-occurrences of ozone and temperature extremes in the northeast U.S. The CMAM and MRI simulations (Figures S7a, b) do better in this regard, although the CMAM simulation does not capture the maximum in the southeastern U.S. There are also discrepancies between the simulations in the midwestern U.S. Student's $t$-test suggests that the GCM2000 fails to capture the extreme dependence between temperature and ozone at the 95% level; however, in the CESM1 REFC1SD simulation we cannot reject with a 95% confidence interval the null hypothesis that the simulated and measured $\varphi$ are the same. Consistent with measurements, all simulations using the CESM1 (including the GCM2100 simulation) show the maximum co-occurrence of temperature and ozone maximum in the southeastern U.S. It is in this region that the co-occurrence of ozone and temperature maxima increase in the future (Figure 9d).

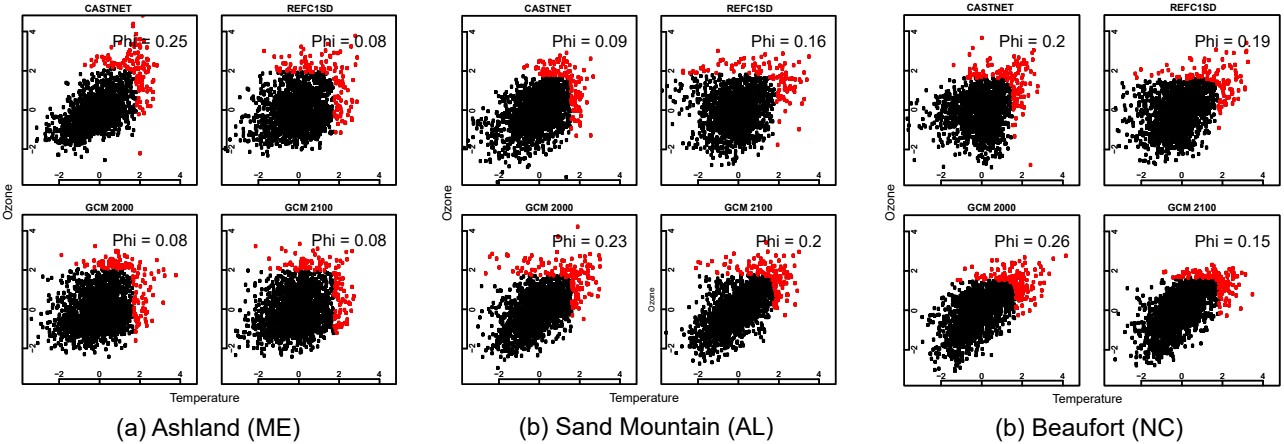

(a) Ashland (ME)    (b) Sand Mountain (AL)    (b) Beaufort (NC)

**Figure 8.** [Deseasonalized Data] The scatter plots of temperature and ozone from selected CASTNET sites (or the corresponding grid point for the CESM1 REFC1SD, the GCM200 or the GCM2100 simulation). (a) Ashland (ME), (b) Sand Mountain (AL), (c) Beaufort (NC). Extreme points picked (red) if the transformed points by using the ranks method are outside the unit circle (see the ranks method in B).

## 5    Discussion

All the CCMI REFC1SD simulations (the CESM1 REFCS1D, CHASER, CMAM, MOCAGE and MRI) and the GCM2000 simulation show a fundamental mismatch between the locations where the width of the right hand side (rhs) of the temperature distribution is large and those locations where rhs of the ozone distribution is large. As measured by the difference between
5  the 20-year return temperature and the mean temperature (Figure 5; Figure S4) or the difference between the $90^{th}$ percentile temperature and the mean temperature (Figure 2; Figure S1) the width of the rhs of the temperature distribution is highest in the northern portion of the domain with a southward extension through the midwestern states and into northwestern states. This pattern is consistent with increased temperature variability at higher latitudes (e.g., Deser et al. (2012) and references therein) and a higher temperature variance in the interior of the country due its greater continentality. However, ozone is most sensitive
10  to temperature changes, as measured by the slope of ozone versus temperature, where ozone precursor emissions are large (Pusede et al. (2015)). This is consistent with the fact that the width of the rhs of the ozone distribution is widest (Figure 5, Figure 3, and Figures S2-S5) in the eastern third of the country, where emissions of ozone precursors are generally the largest. Ozone is also most sensitive to high temperatures in the eastern part of the U.S. ((Figure 7, and Figure S7a, b). Geographical differences between the shape of the ozone and temperature distributions over the U.S. impacts the relationship between ozone
15  and temperature extremes.

The response of ozone to changes in temperature is in part determined by the temperature-ozone correlation. Details of the temperature-ozone correlation are different in all simulations (see Figure 6 and Figure S6). Important differences include the location of the region of low correlations in the south central part of the U.S., the relative strength of the correlation in the northeast and southeast U.S. and the pattern of correlations in the western U.S. Thus we might expect the ozone response to

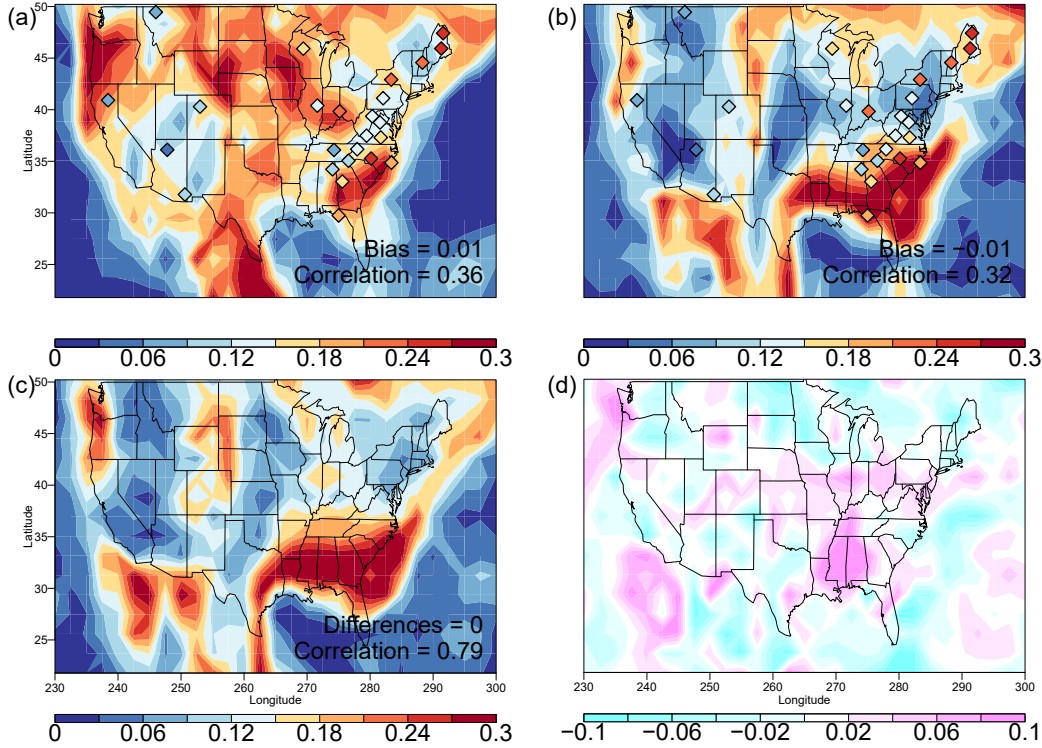

**Figure 9.** [Deseasonalized Data] Areas $\varphi$ from (a) CESM1 REFC1SD simulation (1992-2010), (b) GCM2000 simulation (2006-2025), (c) GCM2100 simulation (2106-2125). Areas $\varphi$ from CASTNET measurements (1992-2011) are shown as filled diamonds in (a) and (b). The bias and correlation in (a), (b) and the difference and correlation in (c) are defined as in Figure 2. (d): Area $\varphi$ from GCM2100 simulation (2106-2125) minus areas $\varphi$ from GCM2000 simulation (2006-2025).

temperature extremes will differ in the different simulations. The CASTNET and AQS measurements (Figure 9 and Figure S6) generally support high temperature-ozone correlations in the Northeast U.S., low correlations along the Gulf Coast and generally high correlations in the Rockies and West coast states (also see Shen et al. (2016)).

Some of the differences between the CESM1 REFC1SD and GCM2000 simulations are likely due to meteorological differences: while the CESM1 REFC1SD simulation is driven with analyzed meteorology, the GCM2000 simulation is driven with model calculated meteorology. In general, the CESM1 REFC1SD simulation captures the measured relation between ozone and temperature better than the GCM2000 simulation over the northeast U.S., although it does not fully capture their strong measured correlation (e.g., see Figure 6). In the Southeast the measured response appears to be generally well simulated in both simulations. Overall the GCM2000 fails to capture the extreme dependence as measured by $\varphi$ between temperature and ozone at the 95% level; in the CESM1 REFC1SD simulation we cannot reject the null hypothesis that the simulated and measured values are the same.

A poor simulation of the Bermuda high in the CGM2000 simulation may be important in explaining some of the differences between the CESM1 REFC1SD and GCM2000 simulations. The position of the Bermuda High strongly impacts ozone distribution over the U.S. (Zhu and Liang (2013)) with the second empirical orthogonal function of ozone variability strongly correlated with the location of the Bermuda High (Shen et al. (2016)). A westward extension of the Bermuda High is corre-

lated with high temperatures and low ozone over the much of the Southeast U.S. (Zhu and Liang (2013)) consistent with the low correlation between ozone and temperature in all the REFC1SD simulations extending northward from the Gulf of Mexico(Figure 6; Figure S6) extending northward from the gulf. We note this region of low correlations in the CMAM and MRI REFC1SD simulations is to the west of that simulated in the CESM1 REFC1SD. In the GCM2000 simulation the Bermuda High is simulated too far to the west (not shown). Consistent with this, a maximum covariance analysis shows that the mode

of variability associated with the Bermuda High is also displaced too far to the west (not shown). Thus, it is likely the pattern of variability associated with the Bermuda High is incorrectly simulated in the GCM2000 simulation. In particular, the GCM simulations do not show a region of low correlation extending northward from the Gulf, but instead a region of low correlation is situated well to the north over Kansas. Zhu and Liang (2013) show the Bermuda High is not well simulated in the majority of GCMs. This has important implications for the simulation of the ozone response to temperature over large sections of the

country.

In this study, we introduce a new spectral method using multivariate extreme value theory to measure extremal dependence between temperature and ozone in both the observations and model simulations. We find through the use of this new metric joint extremes of temperature and ozone occur together up to approximately 35% of the time in a few regions, although on average their joint occurrence is significantly less. Previous studies have used different methodologies to capture the extremal

dependence in measurements between ozone and temperature (Sun et al. (2017), Schnell and Prather (2017), Zhang et al. (2017)). The analysis here uses a somewhat different methodology so it cannot be compared quantitatively with the previous results, but qualitatively the overall patterns are similar to those found previously. The advantage of the spectral method for finding joint extremes of temperature and ozone is that it gives detailed information about the joint extremes and is not restricted to a particular quantile of the distribution. It can be used to forecast joint extremes even out of the range of available samples.

The various model simulations differ in their simulation of $\varphi$ (measuring the joint spectral extremes of ozone and temperature), again suggesting ozone may respond differently to high temperatures in the different simulations. In the central part of the country where the right-hand side of the temperature distribution is particularly wide $\varphi$ is relatively small (Figure 9; Figure S7) in all but the REFC1SD CESM1 simulation (Figure 9; Figure S7). $\varphi$ is high in the CMAM and the MRI simulations in the northeast U.S., but not in the CESM1 REFC1SD or the GCM2000 simulations; on the other hand in the CESM1 REFC1SD,

the GCM2000 and the MRI simulations $\varphi$ is high in the southeast U.S., but not in the CMAM simulation. $\varphi$ is high in the majority of the simulations in the northwestern states.

In general (with some exceptions along the U.S. west coast) the geographical pattern of averaged MDA8 ozone conditioned on daily maximum temperature greater than the 90th percentile $(mean(O_3|T > 90\%) - mean(O_3))$ is qualitatively similar to that of $\varphi$ in the model simulations and the measurements. However, it is important to note that while the measurement sites

used here are sufficiently dense in the eastern U.S. to resolve some of the regional features, they are nowhere dense enough to

resolve regional features in the western two thirds of the country. The CASTNET measurements show the northeast region and some sites in the Southeast have the largest response of ozone to temperature extremes (Figure 7). These same regions have the highest spectral dependence between ozone and temperature extremes (Figure 9) and the highest measured correlations between temperature and ozone conditioned on temperature greater than the $90^{th}$ percentile level (Figure 6). Note, that the CASTNET
measurement sites in the far southeastern part of the country have comparatively small measured variations in relative extreme temperature $(mean(T|T > 90\%) - mean(T))$. Thus, at some sites over the southeastern U.S. the sensitivity of ozone to changes in temperature is relatively large. In contrast, over the northeastern U.S. both the relative variations in the width of the rhs of the MDA8 ozone distribution $(mean(O_3|O_3 > 90\%) - mean(O_3))$ and the rhs of the daily maximum temperature $(mean(T|T > 90\%) - mean(T))$ are relatively high and thus the ozone sensitivity to temperature change is relatively small.

Comparing the GCM2100 and the GCM2000 simulations the mean future temperature increases everywhere in relation to the current climate, although in some places the width of the rhs of the future temperature distribution decreases. The future difference between summertime maximum daily temperatures at the $90^{th}$ percentile minus mean maximum daily temperature increases by up to 20 to 30% compared to the present day $(\Psi(T,T))$ over parts of the southern Mississippi basin extending to the northern Midwest and the Northwest coast (Figure 4). The pattern of this increase bears a striking resemblance to those
locations where measured summertime inter-annual extreme temperatures increase relative to inter-annual increases in the mean (Huybers et al. (2014)), a process linked to drying of the soils (Huybers et al. (2014)). In other parts of the country the relative future increase is small or negative. Note that the increase in $(\Psi(T,T))$ over the lower Mississippi valley occurs in a location strongly impacted by the Bermuda High and thus warrants further investigation. Note also, this increase occurs in those locations where the CESM1 REFC1SD simulation gives a low correlation between ozone and temperature.

To what extent are the relative changes in the future width of the MDA8 ozone distribution determined by the relative future changes in that of the maximum daily temperature distribution? In most locations the future width of the ozone distribution decreases. An exception is in the Midwest where the increase in the future width of the maximum daily temperature distribution is most pronounced. The spatial correlation between $\Psi(T,T)$ and $\Psi(O_3,O_3)$ is significant, but weak, with a correlation coefficient of 0.3. This suggests a weak relationship between changes in the right-hand side of the future MDA8 ozone dis-
tribution and the future maximum daily temperature distribution. Overall, future ozone is less responsive to temperature than present day ozone $(mean(O_3|T > 90\%) - mean(O_3))$ but the effect is small averaged over the continental U.S. (-0.26 ppb) suggesting only relatively modest temperature suppression. The ratio of future sensitivities to temperature compared to present varies regionally ranging from -50% to +50% (Figure 7). Interestingly, ozone does become more responsive to temperature changes in the lower Mississippi valley (Figure 7), in precisely the region that the width of the temperature distribution in-
creases (Figure 4). However, even in this region the rhs of the future ozone distribution does not become significantly wider than its present-day values (Figure 4).

There have been different predictions as to whether climate change increases future ozone extremes with respect to the increase in the mean (e.g., see Sun et al. (2017)). On average the width of the rhs of the future MDA8 ozone distribution increases slightly by 0.26 ppb in the future simulation $(mean(O_3|O_3 > 90\%) - mean(O_3))$ (Figure 2, but also see Figure 5
for the relative change in the 20-year return level), but in many locations the relative width decreases $(\Psi(O_3,O_3)$, Figure 4).

Where it increases, the increase is always less than 20%. Our results generally suggest that the increase in future ozone is primarily due to a shift in the ozone distribution and not due to an increase in ozone at the high end.

## 6   Conclusion

We investigate high temperature and ozone extremes and their joint occurrence over the United States during the summer months (JJA) in measurements and simulations of the present and future climate. Three simulations using the CESM1 with chemistry were analyzed: the CESM1 CCMI reference experiment using specified dynamics (REFC1SD) between 1992-2010, a 25-year present-day simulation branched off the CCMI REFC2 simulation in the year 2000 (GCM2000) and a 25-year future simulation branched off the CCMI REFC2 simulation in 2100 (GCM2100). Distinct from the CCMI REFC2 simulations the emissions and long-lived greenhouse gas distributions (except CO2) are held constant in the GCM2000 and GCM2100 simulations at values representative of the year 2000. In addition, we analyzed the REFC1SD simulation in four additional models with data available at sufficiently high temporal frequency: the CHASER, CMAM, MOCAGE and MRI models. All the CCMI REFC1SD simulations (the CESM1 REFCS1D, CHASER, CMAM, MOCAGE and MRI) and the GCM2000 simulation have a large bias in maximum daily ozone. Scaled ozone biases are 12 and 21 ppb respectively in the CESM1 REFC1SD and GCM2000 simulation. Consistent with many global model simulations the ozone bias is particularly pronounced over the eastern U.S. The simulation of daily maximum temperatures show considerable variability between the various model simulations.

The average global mean daily maximum temperature change between the present-day simulation (GCM2000) and the future simulation (GCM2100) simulation is $2.1°C$, less than the $2.8°C$ difference in the parent CCMI REFC2 simulations. The difference between these set of simulations is most likely attributable to the fact the GCM2100 simulation includes the effect of increased CO2 forcing in the future, but does not account for the impact of projected future aerosol decreases. Thus, in the GCM2100 simulation the relatively large aerosol radiative forcing acts as a buffer against the increased CO2. Over the continental U.S. ozone increases by approximately 2.1 ppb between the GCM2000 simulation and the GCM2100 simulation.

The main conclusions from this study are as follows:

- Five out of six of the simulations analyzed underestimate the width of the measured tail at the high end of the ozone distribution in the present climate, despite the fact that all simulations overestimate the mean ozone. The 20-year return period of ozone minus its mean is underestimated by more than 9 ppb in the GCM2000 simulation evaluated over all CASTNET sites, while in the REFC1SD CESM1 simulation it is underestimated by somewhat more than 3 ppb. The 20-year return period of temperature minus its mean is generally about $2°C$ less than measured in both the GCM2000 and the CESM1 REFC1SD simulations. Despite large biases in mean daily maximum temperature the bias in the width of the rhs of the temperature tails in the CMAM and MRI simulations $(mean(T|T > 90\%) - mean(T))$ is less than $1°C$.

- We propose a new method to measure the joint extremes of temperature and ozone by calculating the spectral density $(\varphi)$ of the joint extremes of ozone and temperature. This measure of the joint extremes is not restricted to a particular

quantile of the distribution, but can be used to forecast joint extremes even out of the range of available samples. While in many areas of the country MDA8 ozone and maximum daily temperature are highly correlated, the correlation is reduced significantly at the higher end of the distributions. Measures of spectral density are everywhere less than about 0.35, so that only about a third of the time at most do extreme temperatures coincide with extremely high ozone. Observations show that $\varphi$ is highest in the northeast U.S. and in the southeast U.S. To some extent this response is consistent with the ozone response to extreme temperatures ($mean(O_3|T > 90\%) - mean(O_3)$). To the extent the measurements are dense enough to define the spectral density geographically, the simulations capture much of the measured pattern.

– In all simulations there is a geographical mismatch between where the rhs of the simulated maximum daily temperature distribution is large and where the rhs of the simulated MDA8 ozone distribution is large. Thus, while ozone concentrations are often correlated with temperature, the regions of high ozone extremes do not necessarily match the regions of high temperature extremes. All things being equal, we might expect the ozone distribution to be a slave to the temperature distribution so that regions of particularly high temperature extremes might also be expected to have particularly high ozone extremes. However, this is not the case. The highest temperature extremes tend to occur in the Midwest while the highest ozone extremes tend to occur in the eastern part of the country. Regions with high ozone precursor emissions are known to increase the ozone-temperature slope making ozone in regions with high precursor emissions sensitive to smaller temperature variations. Other complicating factors such as the importance of biogenic emissions or regional meteorological differences may also complicate the distributional relation between ozone and temperature.

– The various model simulations show some rather pronounced differences in the ozone-temperature relationship. These differences suggest that ozone will respond rather differently to temperature changes in the various simulations. Differences between the CESM1 REFC1SD and the GCM2000 simulation can be attributed in part due to differences in the meteorology. The response of the REFC1SD simulation is qualitatively better than that of the GCM2000 simulation. We hypothesize that the differences in these simulations are meteorologically induced and may, at least in part be attributed to a poor simulation of the Bermuda High in the GCM2000 simulation.

– In the future climate the ozone and temperatures distributions shift to the right. Our results generally suggest that the increase in both future temperature and future ozone is primarily due to a shift in the distributions, not to an increase in the extremes. Overall, the rhs of the temperature distribution increases slightly, with the largest increase, when evaluated at the CASTNET sites, in the Midwest. In some locations the increase in the rhs of the temperature distribution approaches 30%, in other locations it decreases up to 10%. The pattern of increase is what might be expected from soil-moisture feedbacks. On average the width of the rhs of the future ozone distribution increases slightly by 0.26 ppb in the future simulation ($mean(O_3|O_3 > 90\%) - mean(O_3)$), with regional increases up to 20% and decreases up to 10% over parts of the northeastern U.S. and much of the western two thirds of the country. At CASTNET sites increases in the 20-year return period minus the mean are only found on average in the Midwest. The correlation between relative changes in the high end of the future temperature distribution ($\Psi(T,T)$) and the ozone distribution ($\Psi(O_3,O_3)$) is 0.3, relatively small but still significant. Thus an increase in the rhs of the future temperature distribution may have some impact on the future

ozone distribution. However, the correlation is weak suggesting other complicating factors. It is possible that a stronger relationship would emerge from longer model simulations. In any case the region where the rhs of the temperature distribution increases does not correspond to the region where the width of the rhs of the ozone distribution is large. Elsewhere in the world, perhaps in regions with strong soil-moisture feedbacks and high emissions of ozone precursors, a future amplification in the future temperature distribution would have more dramatic impacts on future ozone extremes.

## Appendix A: Univariate regular variation

To understand the basic characteristics of extreme distributions, we should introduce the notion of *regular variation*. A regularly varying function is a function whose behavior at infinity follows a power law function. That is, a regularly varying function with an index $\alpha$ can be explained by

$$\lim_{t \to \infty} \frac{F(tx)}{F(t)} = x^{\alpha}, \tag{A1}$$

for all $x > 0$.

Regularly varying functions are studied in many fields and one of the applications that we will use here is to estimate the tail indices $\alpha$ of extreme ozone and extreme temperature distributions in order to estimate $N$-year return levels of those variables. Alternatively, we can fit the ozone or temperature distributions to the Generalized Pareto Distribution (GPD) and estimate the shape parameters which are equivalent to the reciprocal of the tail indices (shape = $\alpha^{-1}$) if the shape parameter is positive. Phalitnonkiat et al. (2016) suggests a procedure to estimate shape parameters using a combination of Hill estimators and Maximum Likelihood Estimators (MLE).

## Appendix B: Ranks method

Let us consider 2-dimensional random vectors $(X, Y)$. When the tail part of $X$'s distribution and the tail part of $Y$'s distribution are independent, we would expect that $X$ and $Y$ are unlikely to yield extreme values at the same time, and vice versa. This observation suggests that when we plot only extreme points from $(X, Y)$, the points would appear to be around the axes if $X$ and $Y$ are extreme independent, and vice versa. This is actually true in higher dimensions as well. However, the tool described above for measuring the dependence between variables only applies to variables with the same marginal tail indices.

Among the methods suggested by Resnick (2007), we use a transformation that essentially normalizes the tail indices of all components to 1 without calculating or estimating the tail indices $\alpha_j$ for each $j = 1, ..., d$. This method is called the *Ranks methods*. The major benefit from this method is that we can avoid the marginal tail index estimation which reduces numerical errors; however, the drawback is that the transformation itself destroys the iid (independent and identically distributed) property of the data and makes it more complicated to obtain asymptotic distributions, see Einmahl et al. (2001). The method can be done as follows.

Let $\mathbf{X}_i = (X_i^{(1)}, ..., X_i^{(d)}), i = 1, ..., n$ be $d-$dimensional vectors. Denote the *rank* of $X_i^{(j)}$ by

$$r_i^{(j)} := \sum_{m=1}^{n} 1_{[X_m^{(j)} \geq X_i^{(j)}]}. \tag{B1}$$

For a fixed $k > 0$ and for each $i = 1, ..., n$ we transform $\mathbf{X}_i$ into a rank vector by

$$\left( X_i^{(1)}, ..., X_i^{(d)} \right) \mapsto \left( \frac{k}{r_i^{(1)}}, ..., \frac{k}{r_i^{(d)}} \right). \tag{B2}$$

We consider a point $\left( X_i^{(1)}, ..., X_i^{(d)} \right)$ as jointly extreme if $\left\| \left( \frac{k}{r_i^{(1)}}, ..., \frac{k}{r_i^{(d)}} \right) \right\| > 1$, where $||\cdot||$ is a norm in $\mathbb{R}^d$. In this case, we use the $L^2$-norm or least squares. We use the transformed vectors to estimate the spectral measure (or angular measure in the case of 2-dimensional vectors).

## Appendix C: Estimating spectral measure

To estimate the spectral measure from the data in 2-dimensional polar coordinates, we measure the angles between the transformed points $\left( \frac{k}{r_i^{(1)}}, \frac{k}{r_i^{(2)}} \right)$ and the $x$-axis. That is, we can estimate the empirical measure $\hat{S}$ by

$$\hat{S}(A) = \frac{\# \text{ of extreme points with angles in } A}{\# \text{ of extreme points}}, \tag{C1}$$

where $A$ is a set or an interval. Note that this can be extended to higher dimensions in a similar way.

We may notice that the choice of $k$ has a major role on how we categorize extreme points. The higher $k$ is, the more points would lie outside the unit circle, and hence, the more extreme points. We use the procedure from Nguyen and Samorodnitsky (2013) to estimate $k$.

Since the angular measure is normalized (i.e., the area under curve from 0 to $\frac{\pi}{2}$ is 1), we can only consider the area of the 'middle' part, which we define to be between $\frac{\pi}{8}$ and $\frac{3\pi}{8}$. Denote this amount by $\varphi$:

$$\varphi := \hat{S}\left( \left[ \frac{\pi}{8}, \frac{3\pi}{8} \right] \right) \approx \text{area} \left[ \frac{\pi}{8}, \frac{3\pi}{8} \right], \tag{C2}$$

where the area is defined in a notion of kernel density estimation from the angular measure.

*Author contributions.*

*Competing interests.* The authors declare that they have no conflict of interest.

*Disclaimer.*

*Acknowledgements.* We would like to thank the three anonymous referees whose careful reading considerably improved this paper. This research was made possible by EPA Award RD-83520501 and NSF award number 1608775. Its contents are solely the responsibility of the grantee and do not necessarily represent the official views of the USEPA. The CESM project is supported by the National Science Foundation and the Office of Science (BER) of the U. S. Department of Energy. The National Center for Atmospheric Research is funded by the National

5   Science Foundation. In addition, we acknowledge the joint WCRP SPARC/IGAC Chemistry-Climate Model Initiative (CCMI) for organizing and coordinating the model data analysis activity, and the British Atmospheric Data Centre (BADC) for collecting and archiving the CCMI model output.

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

**Table 1.** Details and descriptions for each model.

| Simulation (Years) | GHG[1] forcing | Emissions | SST[2] and sea ice | Meteorology |
|---|---|---|---|---|
| CESM-REFC1SD (1992-2010) | CMIP5[3] (updated until 2010) | Anthropogenic and biomass burning emission: MACCity[4] Biogenic emissions: MEGAN2[5] | HadISST2[6] | MERRA[7] |
| GCM2000 (2006-2025) | $CO_2 = 369$ ppm. Other GHG from REFC1SD. | Anthropogenic and biomass burning from AR5[8]. Biogenic emissions: Monthly values from MEGAN2 for 2000 | Online | Online |
| GCM2100 (2106-2125) | $CO_2 = 669$ ppm. Other GHG as in GCM2000. | GCM2000 | Online | Online |

[1] Greenhouse gas. [2] Sea surface temperature. [3] Coupled Model Intercomparison Project. [4] Granier et al. (2011). [5] Guenther et al. (2012). [6] Hadley Center Sea Ice and Sea Surface Temperature data set (Titchner and Rayner (2014)). [7] Modern Era Retrospective-analysis for Research and Applications (Rienecker et al. (2011)). [8] Assessment Report 5 (Eyring et al. (2013a)).

**Table 2.** [Rescaled Data] MDA8 ozone averages (ppb) and daily maximum temperature averages ($^{\circ}C$) in different regions over the U.S. from CASTNET data and corresponding grid points from the CESM1 REFC1SD, the GCM2000 and the GCM2100 simulations. The averages are calculated from MDA8 (for ozone) and daily maximum (for temperature). Standard deviations (sd) are calculated between the stations in each region. The averages of ozone and temperature from each region are reported in italics (including all continental grid points in each box in Fig. 2a). The italics under 'All' are averages of all points over the continental U.S.

| Region | CASTNET Ozone (sd) | CASTNET Temp (sd) | REFC1SD Ozone (sd) | REFC1SD Temp (sd) | GCM 2000 Ozone (sd) | GCM 2000 Temp (sd) | GCM 2100 Ozone (sd) | GCM 2100 Temp (sd) |
|---|---|---|---|---|---|---|---|---|
| Northeast | 45.46 | 23.01 | 59.48 | 19.81 | 66.46 | 23.39 | 71.03 | 25.98 |
|  | (9.9) | (1.8) | (7.59) | (1.4) | (11.08) | (1.53) | (11.06) | (1.33) |
|  | | | *60.33* | *20.27* | *67.48* | *23.11* | *72.08* | *25.75* |
| Southeast | 52.71 | 25.38 | 62.31 | 24.86 | 72.62 | 27.46 | 75.34 | 29.36 |
|  | (4.99) | (3.17) | (8.63) | (1.77) | (9.81) | (0.9) | (10.68) | (0.59) |
|  | | | *61.56* | *24.90* | *72.27* | *27.59* | *74.88* | *29.5* |
| Midwest | 44.67 | 25.91 | 65.1 | 23.68 | 77.59 | 27.67 | 80.95 | 30.16 |
|  | (8.22) | (3.93) | (8.11) | (1.53) | (7.8) | (1.42) | (7.24) | (1.35) |
|  | | | *64.63* | *23.61* | *77.92* | *27.50* | *81.67* | *29.97* |
| West | 51.29 | 22.27 | 60.47 | 21.81 | 66.15 | 26.62 | 67.1 | 29.12 |
|  | (7.52) | (4.37) | (9.46) | (5.07) | (6.92) | (4.35) | (6.47) | (3.81) |
|  | | | *61.39* | *22.30* | *67.61* | *26.59* | *68.68* | *29.19* |
| All | 49.72 | 24.25 | 61.6 | 22.9 | 70.45 | 26.34 | 73.33 | 28.6 |
|  | (7.63) | (3.39) | (8.16) | (3.27) | (9.75) | (2.67) | (10.14) | (2.33) |
|  | | | *54.28* | *22.76* | *61.65* | *26.16* | *63.72* | *28.59* |

**Table 3.** [Rescaled Data] Twenty-year return levels for MDA8 ozone (ppb) and daily maximum temperature ($^\circ C$) (first and third columns) at the CASTNET sites and for the CESM1 REFC1SD simulation, the GCM2000 simulation and the GCM2100 simulation. The models are sampled only at the CASTNET stations. Twenty-year return levels ozone and temperature minus their averages (second and fourth columns).

| Model | Region | Ozone [ppb] | | Temperature [$^\circ C$] | |
|---|---|---|---|---|---|
| | | 20-year return | Minus mean | 20-year return | Minus mean |
| CASTNET | Northeast | 86.37 | 40.95 | 32.19 | 9.2 |
| | Southeast | 88.83 | 36.28 | 31.77 | 6.42 |
| | Midwest | 80.3 | 35.66 | 33.47 | 7.57 |
| | West | 73.15 | 21.9 | 30.63 | 8.42 |
| | All | 84.08 | 34.45 | 31.85 | 7.63 |
| REFC1SD | Northeast | 95.39 | 35.91 | 25.81 | 6.01 |
| | Southeast | 94.9 | 32.59 | 29.54 | 4.67 |
| | Midwest | 97.98 | 32.88 | 29.93 | 6.24 |
| | West | 81.57 | 21.09 | 28.08 | 6.27 |
| | All | 92.72 | 31.12 | 28.4 | 5.5 |
| GCM 2000 | Northeast | 96.77 | 30.3 | 29.53 | 6.14 |
| | Southeast | 98.96 | 26.34 | 32.07 | 4.61 |
| | Midwest | 102.61 | 25.02 | 34.71 | 7.04 |
| | West | 81.93 | 15.77 | 31.97 | 5.34 |
| | All | 95.46 | 25.02 | 31.75 | 5.41 |
| GCM 2100 | Northeast | 101.33 | 30.3 | 32.38 | 6.4 |
| | Southeast | 100.78 | 25.44 | 34.53 | 5.16 |
| | Midwest | 107.66 | 26.71 | 38.18 | 8.02 |
| | West | 82.42 | 15.32 | 34.92 | 5.81 |
| | All | 98.06 | 24.74 | 34.53 | 5.93 |