# Peer review of "Extremal Dependence between Temperature and Ozone over the Continental U.S."

_Atmospheric Chemistry and Physics, 2017_

## Referee Comment (RC1) · Anonymous Referee #1 · 21 Dec 2017

General comments:

This manuscript uses the results from 3 models runs, in addition to measurement data, in order to investigate the relationships between ozone and temperature extremes. The authors present a new methodology to evaluate the spectral dependence of ozone and temperature extremes.

The manuscript is well written and presents some interesting results. However, I find the manuscript to be lacking in explanation of how the results could be of use to the climate modelling community. In many cases, there are weak correlations between ozone and temperature extremes, but the authors do not explain the significance of these results. It is not clear whether the authors are proposing that weak correlations are due to inaccuracies in the model, or whether the weak correlations mean that

ozone extremes are unlikely be significantly affected by the increasing temperatures associated with climate change. To enhance the readability of the article, I recommend that the following specific comments be addressed.

Specific comments:

Abstract: I suggest adding a concluding sentence to emphasize how the results of this paper are useful to the climate modelling community at large.

Introduction: The first sentence about climate change ozone penalty is an odd choice for the opening sentence of the paper, as the rest of the paper doesn't mention ozone penalty again. I suggest that the introduction be reorganized so that it starts with general background information about the relationship between ozone and temperature, followed by a discussion of how air quality and temperature are expected to change in the future, etc.

Page 2, line 3: Need to clarify that the emission of precursors is the key driving factor for future pollutant levels.

Page 2, line 12: The increase of 2-8 ppbv ozone per degree of Celsius seems rather high. It seems to me that Brown-Steiner et al. (2015) estimate the ozone-temperature relationship to be 0-6 ppb per degree K, not 2-8. Several other studies have lower estimates.

Page 2, line 22: This sentence is confusing as written because how can an increase in return period (which is measured in years) be similar to an increase in temperature (measured in degrees)? Please clarify.

Page 2, line 23: Seneviratne et al. (2012) use the results of CMIP3, not CMIP5.

Page 3, line 7: It would be useful to have a bit more information about how the stations were chosen (e.g. do the sites have to have data for every year in the 1992-2005 period? What percentage of the data can be missing?)

Page 3, line 16: Please clarify where the CO2 concentrations in the GCM come from.

Page 7, line 17: This appears to be 2 ppb, not 3 ppb.

Page 10, line 5: Please clarify how the return level analysis is different from looking at the difference between the 90th percentile and the average, so that the reader can better understand why both of these analyses were used. The results of the return level analysis seem to be consistent with the difference between the 90th percentile and the average, so the benefit of including the return level analysis is unclear.

Page 10, line 15: Are there enough measurement stations in the West and Midwest to be able to draw any meaningful conclusions?

Page 15, line 4: Why were these 3 CASTNET sites chosen?

Page 19, line 10: The Northwest and Southeast have the best correlations, but isn't this partly due to the fact that there are more measurement stations in these areas?

Page 19, line 17: It is misleading to say that there is a strong dependence of ozone upon temperature, because, as your results demonstrate, the relationship between ozone and temperature is complex.

Page 21: I feel that the conclusion is lacking an explanation of how the results of this paper are useful to the climate modelling community. On line 7, for example, what conclusions can you draw from this geographical mismatch? On line 21, you state that the correlation is 0.3: what does this correlation mean for modellers and for future studies?

Technical corrections:

Page 1, line 3: I think it's worth specifying that you are talking about "tropospheric" ozone.

Page 2, line 23: Spell out the acronym CMIP.

Page 3, line 10: Spell out the acronym MERRA.

Page 5, line 11: Missing the word "Appendix" before the letter C

Page 6, caption of Figure 1: Subject (Examples) does not agree with verb (shows)

Page 7, line 23: Please fix this sentence so that it doesn't contain two occurrences of the word "while"

Page 7, line 32: "Northwards" (a direction) should not be capitalized, whereas d"Midwest" (a proper noun) should be capitalized. Similar capitalization errors are found throughout the document.

Page 17, line 2: "in interior" should be "in the interior"

Page 22, line 2: Spell out the acronym MLE.

Page 22, line 12: Spell out the acronym iid.

Page 22, line 16: What is Rd?

Page 22, line 17: It would be helpful to note that $L^2$ is the least squares regression.

Page 22, line 22: What is A?

Throughout: When previous studies, of the type "Author et al.", are used as the subject of a sentence, the verb that follows should be conjugated in the plural form. The paper currently has a mixture of both singular and plural verb forms after "Author et al."
* * *

---

## Referee Comment (RC2) · Anonymous Referee #3 · 31 Dec 2017

The authors present an interesting study on the dependence of ozone and temperature extremes in both observations and model simulations. Particularly the methodology is innovative and expands beyond the standard tools in the field. However, as varies studies addressing this complex topic before, the present work is not able to provide clear answers and raises several questions while answering others.

Thus I recommend revisions and publication in ACP after the authors have addressed the comments below.

Major Comments:

1. The authors provide an elegant methodology to minimize the influence of year-to-year variability and seasonal effects. At the same time this procedure comes at

the cost of not considering all/many extremes on record. I am wondering about the effect the omission number a (here a=10) has on the results. How robust are the presented findings for values of a=5, a=15, a=1? Also I am curious if an approach omitting a certain number of extremes is more robust than more standard de-clustering approaches? It would be great if the authors could include a statement on this (or maybe an additional appendix) in the revised manuscript.

2. CAM4-chem is only one of the models contributing to the CCMI effort. In this light it would be great to see how other models that performed the REFC1SD experiment agree with observations. That said, I am not asking the authors to provide an in depth analysis across various models. It would be though interesting and provide important context for the community if the bias found for the model considered is 'common' in magnitude and sign across models and coherent across regions.

3. The authors use the set of CASTNET sites for comparison with model data, however we do not learn about which site selection criteria have been applied.

4. CASTNET data seems a robust choice to evaluate a global model for the US domain. However given the complications in the spatial correlation patterns of temperature and ozone I wonder if supplementing the observational data set with a suite of selected AQS sites might help. In the current analysis only 5 sites are used to cover the entire West, which raises robustness concerns.

5. The authors choose three sites (Ashland, Sand Mountain, Beaufort) to illustrate ozone temperature correlations. How have those stations been chosen from the CAST-NET set, and would it be not more intriguing to pool stations for the spatial domains indicated in Fig. 2a?

6. The correlations reported are relatively low, which is acknowledged by the authors. I am wondering though if it would not be worthwhile to report explained variance throughout the manuscript instead of correlation coefficients.

7. The key conclusions of the manuscript should be highlighted and also included in the abstract. Right now the language is a little vague regarding the significance and robustness of the overall extremal dependence. Despite all caveats raised in the discussion section, the study suggests a weak but robust relationship between ozone and temperature extremes. At the same time the spatial mismatch between regions where high ozone and high temperature extremes occur is of relevance and will motivate future modelling work.

Technical Comments:

Title: "Extremal dependence . . . " might not be very accessible for non-statisticians

---

## Referee Comment (RC3) · Anonymous Referee #2 · 5 Jan 2018

This paper presents an interesting study that examines the relationship between temperature and ozone extremes in measurements as compared 3 chemistry-climate model simulations. They develop a new metric to measure the joint extremal dependence of ozone and temperature by evaluating the spectral dependence of their extremes. The paper is thorough and well-written and contains interesting new results, thus I recommend publication following minor revision.

Page 2, lines 3-6: Make clear which regions are being referred to.

Page 2, line 15: temperature dependent emissions of both anthropogenic and biogenic in origin.

Page 3, line 5: Why were these sites selected? - their operational length?

[Figure]

Page 4, line 10: How does the choice of a (= 10) affect the robustness of the results?

Page 5, line 12: Appendix C

Page 7, line 7: Should this be "does not change . . ."?

Page 7, line 21: Is this due to biases in the 90th or mean?

Page 7, line 24: the 2nd "while" isn't needed

Page 10, line 22: This sentence seems to be missing something. . . The greatest simulated differences?

Page 11, line 20: Could EPA AQS data be added to fill some of these gaps?

Page 10, lines 10-20: A discussion on comparing the performance of REFC1SD to GCM2000 would be useful here and elsewhere, where appropriate – namely, the effect of having nudged meteorology rather than a free running simulation that may not simulate the synoptic conditions conducive to ozone and/or temperature extremes.

Page 15, line 4: How are these sites chosen?

───────────────────────

---

## Author Comment (AC1) · 13 Jun 2018

We wish to thank all the referees for their helpful comments. The comments have resulted in a considerable improvement to the paper. We have added the following main changes to the paper: (i) we have rewritten the introduction at the recommendation of referee #1. As a result we have moved a discussion of the various metrics that have been used to analyze the extremal dependence of ozone and temperature from the discussion section to the introduction. Consequently, the new discussion section is also considerably revised. (ii) At the recommendation of referee #3 we have included results from other CCMI models when available. The results are graphically analyzed in the supplement but referred to throughout the paper.

Specific referee comments (comments in red, our reply in black).

Referee #1

The manuscript is well written and presents some interesting results. However, I find the manuscript to be lacking in explanation of how the results could be of use to the climate modelling community. In many cases, there are weak correlations between ozone and temperature extremes, but the authors do not explain the significance of these results. It is not clear whether the authors are proposing that weak correlations are due to inaccuracies in the model, or whether the weak correlations mean that ozone extremes are unlikely be significantly affected by the increasing temperatures associated with climate change.

Hopefully, the following additions to the paper will clarify the significance of the results to the climate modeling community.

We have rewritten the last part of the abstract to read:

"Measures of spectral density are everywhere less than about 0.3, suggesting that at most only about a third of the time do very high temperatures coincide with very high ozone. Two regions of the U.S. have the strongest measured extreme dependence of ozone and temperature: the Northeast and the Southeast. The simulated future increase in temperature and ozone is primarily due to a shift in their distributions, not to an increase in their extremes. The locations where the right-hand side of the temperature distribution does increase (by up to 30%) are consistent with locations where soil-moisture feedback may be expected. Future changes in the right-hand side of the ozone distribution range regionally between +20% and -10%. The location of future increases in the high end tail of the ozone distribution are weakly related to those of temperature with a correlation of 0.3. However, the regions where the temperature extremes increase are not located where the extremes in ozone are large, suggesting a muted ozone response."

We also discuss the relationship between regions with large ozone and temperature extremes more thoroughly in the conclusion:

"All things being equal, we might expect the ozone distribution to be a slave to the temperature distribution so that regions of particularly high temperature extremes would be expected to have particularly high ozone extremes. However, this is not the case. High temperature extremes tend to occur in the Midwest while the highest ozone extremes tend to occur in the eastern part of the

country. Regions with high ozone precursor emissions are known to increase the ozone-temperature slope so that regions with high precursor ozone emissions are particularly sensitive to temperature variations. Other complicating factors such as the importance of biogenic emissions or regional meteorological differences may also complicate the distributional relation between ozone and temperature."

We also clarify what a correlation of 0.3 implies between the increases in the right-hand side of temperature and ozone in the conclusions:

"In the future climate the ozone and temperatures distributions shift to the right. Our results generally suggest that the increase in both future temperature and future ozone is primarily due to a shift in the distributions, not to an increase in the extremes. Overall, the rhs of the temperature distribution increases slightly, with the largest increase, when evaluated at the CASTNET sites, in the Midwest. In some locations the increase in the rhs of the temperature distribution (mean(T|T>90%) - mean(T))approaches 30%, in other locations it decreases up to 10%. The pattern of increase is what might be expected from soil-moisture feedbacks. On average the width of the rhs of the future ozone distribution increases slightly by 0.26 ppb in the future simulation (mean(O3|O3>90%) - mean(O_3)), with regional increases up to 20% and decreases up to 10% over parts of the northeastern U.S. and much of the western two thirds of the country. When evaluated at CASTNET locations increases in the 20-year return period minus the mean are generally confined to the Midwest. The correlation between relative changes in the high end of the future temperature distribution (PSI(T,T)) and the ozone distribution (PSI(O3,O3)) is 0.3, relatively small but still significant. Thus an increase in the rhs of the future temperature distribution may have some impact on the future ozone distribution. However, the correlation is weak suggesting other complicating factors. It is possible that a stronger relationship would emerge from longer model simulations. In any case the region where the rhs of the temperature distribution increases does not correspond to the region where the ozone extremes are relatively high. Elsewhere in the world, perhaps in regions with strong soil-moisture feedbacks and high emissions of ozone precursor emissions, a future amplification in the future temperature distribution would have more dramatic impacts on future ozone extremes."

To enhance the readability of the article, I recommend that the following specific comments be addressed.

Specific comments:
Abstract: I suggest adding a concluding sentence to emphasize how the results of this paper are useful to the climate modelling community at large.

We have added the sentences as described above which we think will be helpful to the climate and pollution modeling community at large.

Introduction: The first sentence about climate change ozone penalty is an odd choice for the opening sentence of the paper, as the rest of the paper doesn't mention ozone penalty again. I suggest that the introduction be reorganized so that it starts with general background information about the relationship between ozone and temperature,

followed by a discussion of how air quality and temperature are expected to change in the future, etc.

We have rewritten the introduction to take out references to the ozone penalty.

Page 2, line 3: Need to clarify that the emission of precursors is the key driving factor for future pollutant levels.

The paragraph has been largely rewritten. This comment is no longer pertinent to the revised paper.

Page 2, line 12: The increase of 2-8 ppbv ozone per degree of Celsius seems rather high. It seems to me that Brown-Steiner et al. (2015) estimate the ozone-temperature relationship to be 0-6 ppb per degree K, not 2-8. Several other studies have lower estimates.

Corrected. Thank you.

Page 2, line 22: This sentence is confusing as written because how can an increase in return period (which is measured in years) be similar to an increase in temperature (measured in degrees)? Please clarify.

We've reworded as follows:
"Globally the future increase in extreme temperatures (temperatures at the 20-year return period) in CMIP3 are similar to the increase in mean temperature"

Page 2, line 23: Seneviratne et al. (2012) use the results of CMIP3, not CMIP5.

Thank you.

Page 3, line 7: It would be useful to have a bit more information about how the stations were chosen (e.g. do the sites have to have data for every year in the 1992-2005 period? What percentage of the data can be missing?)

We've added the following:

"Hourly ozone and temperature data are taken from 23 CASTNET stations with a nearly continuous data record during the study period of 1992–2013 for the months of June, July, and August (92 days each summer). In addition, to enhance the data record, we included two additional stations (Beufort NC, and Lassen Volcanic CA) where the first 2 years or 3 years of data were missing, respectively. See Figure 2 for station locations."

Page 3, line 16: Please clarify where the CO2 concentrations in the GCM come from.

We've added the following:

"In the present-day simulation (the GCM2000 simulation) the CO2 concentration is specified at 369 ppm, representative of the year 2000; in the future simulation (the GCM2100 simulation) the CO2 concentration is specified at 669 ppm, representative of the 2100 concentration of CO2 in the representative concentration pathway 6 (RCP6)"

Page 7, line 17: This appears to be 2 ppb, not 3 ppb.

Fixed.

Page 10, line 5: Please clarify how the return level analysis is different from looking at the difference between the 90th percentile and the average, so that the reader can better understand why both of these analyses were used. The results of the return level analysis seem to be consistent with the difference between the 90th percentile and the average, so the benefit of including the return level analysis is unclear.

The reviewer is correct that the two methodologies are in fact two different ways of looking at the same thing (see below). We decided to present both methodologies as different readers may prefer return periods vs quantiles, and for completeness we give both quantities. It may also be useful to some readers to have the 20-year return period specified. Note also that the 20-year return level is significantly further to the right of the $90^{th}$ quantile.

We have reworded the sentence as follows:

"As an alternative way of viewing the data we also present the 20-year return levels to describe the marginal extremes. For a stationary independent series the return level is simply related to the value at a given percentile of the distribution. We note, however, that the 20-year return level represents a value considerably further out on the high-end of the distribution than the $90^{th}$ percentile (compare Figures 2b,d,f with Figures 5b,d, f for temperature and Figures 3b,d,f and Figure 5a,c,e for ozone). "

Technical explanation:

For a stationary independent series $\{X_i\}$ with marginal distribution F, the average return of level x is $R(x)=1/(1-F(x))$. It is common to specify the return period R and find the corresponding level x(R), e.g., the 50-year return level is the solution of

$R=50=1/(1-F(x(R)))$ for series with time step of a year.

The 90% level $x_{90}$ is given by $F(x_{90})=0.9$. Its average return is $1/(1-F(x_{90}))$. Both $x_{90}$ and x(R) are in the right tail of F so that they are likely to relate in a similar way to the mean of F.

Thus the two analysis are equivalent in the sense that $x_{90}$ coincides with x(R) if $1-1/R=0.9$.

Page 10, line 15: Are there enough measurement stations in the West and Midwest to be able to draw any meaningful conclusions?

The reviewer is correct: there are probably not enough measurements to make a strong conclusion. We've eliminated the reference to the west and mid-west.

Page 15, line 4: Why were these 3 CASTNET sites chosen?

We've included the following in the paper:

"These sites are selected to show a range of behavior in measured and simulated $\varphi$: at one site measured $\varphi$ is larger than simulated (Ashland, Me), at one site it is less than that simulated (Sand Mountain, Al), and at one site the measured and simulated values are about the same (Beaufort, NC)"

Page 19, line 10: The Northwest and Southeast have the best correlations, but isn't this partly due to the fact that there are more measurement stations in these areas?

We include the following sentence in the revised version prior to page 19, line 10:

"However, it is important to note that while the measurement sites used here are sufficiently dense in the Eastern U.S. to resolve some of the regional features, they are nowhere dense enough to resolve the western two thirds of the country."

Page 19, line 17: It is misleading to say that there is a strong dependence of ozone upon temperature, because, as your results demonstrate, the relationship between ozone and temperature is complex.

Thanks. We eliminated this from the discussion.

Page 21: I feel that the conclusion is lacking an explanation of how the results of this paper are useful to the climate modelling community. On line 7, for example, what conclusions can you draw from this geographical mismatch? On line 21, you state that the correlation is 0.3: what does this correlation mean for modellers and for future studies?

We have tried to clarify some of the implications in the abstract and the conclusion. Specifics are given at the beginning of this reply.

Technical corrections:
Page 1, line 3: I think it's worth specifying that you are talking about "tropospheric" ozone.

We have now specified tropospheric surface ozone.

Page 2, line 23: Spell out the acronym CMIP.

Corrected.

Page 3, line 10: Spell out the acronym MERRA.

Done.

Page 5, line 11: Missing the word "Appendix" before the letter C

Done.

Page 6, caption of Figure 1: Subject (Examples) does not agree with verb (shows)

Done.

Page 7, line 23: Please fix this sentence so that it doesn't contain two occurrences of the word "while"

Fixed.

Page 7, line 32: "Northwards" (a direction) should not be capitalized, whereas "Midwest" (a proper noun) should be capitalized. Similar capitalization errors are found throughout the document.

Fixed throughout

Page 17, line 2: "in interior" should be "in the interior"

Fixed.

Page 22, line 2: Spell out the acronym MLE.

Maximum Likelihood Estimators (fixed).

Page 22, line 12: Spell out the acronym iid.

independent and identically distributed (fixed)

Page 22, line 16: What is Rd?

I think the reviewer is referring to $r_i^{(d)}$, the rank of a point. It is defined in equation (B2).

Page 22, line 17: It would be helpful to note that L2 is the least squares regression.

Noted in the text

A is a set or an interval. Corrected in the text.

Throughout: When previous studies, of the type "Author et al.", are used as the subject of a sentence, the verb that follows should be conjugated in the plural form. The paper currently has a mixture of both singular and plural verb forms after "Author et al."

Fixed.

Referee #2

Page 2, lines 3-6: Make clear which regions are being referred to.

We have rewritten the introduction at the recommendation of Referee #1. This paragraph has been eliminated.

Page 2, line 15: temperature dependent emissions of both anthropogenic and biogenic in origin.

We have also eliminated this line.

Page 3, line 5: Why were these sites selected? - their operational length?

We have reworded the text to explicitly state how these sites were selected.

"Hourly measured ozone and temperature data are taken from 23 CASTNET (Clean Air Status and Trends Network) stations with a nearly continuous data record during the period from 1992-2013 for the months of June, July, and August (92 days each summer). In addition, to enhance the data record, we included two additional stations (Beufort NC, and Lassen Volcanic CA) where the first 2 years or 3 years of data were missing, respectively. See Figure 2 for station locations. CASTNET sites are situated to sample regional ozone concentrations so as to minimize the more local impact of urban areas."

Page 4, line 10: How does the choice of a (= 10) affect the robustness of the results?

"We picked the number 10 originally as we wanted to preserve the extremes at approximately 10% of the points. As we have 92 days of data each summer (for JJA) we excluded 10 points from the analysis. As a sensitivity test we have omitted a different number of extreme points (5, 10 or 15) at a number of measurement sites to determine the difference in average ozone for each summer. These differences are on the order of 1 ppb, small compared to the data average"
 (see figure 1 below).

We have also included the sentence in the text:
"Sensitivity tests at a number of stations suggest the result is not sensitive to $a$"

The text is correct. The ranks method is given in appendix B.

Thank you for catching this . Yes it should be "does not change"

The point we are trying to make here is that the width of the high end of the ozone distribution is biased low compared to the measurements.

We have reworded this to:
"Despite the simulated positive bias in average ozone, the simulated difference between the 90[th] percentile and average ozone is biased low in the CESM1 simulations (Figures Figure 3b, d) with average biases of -0.79 and -4.28 ppb in the CESM1-sdREFC1SD and GCM2000 simulations, respectively. Thus the CESM1 simulations underestimate the width of the high end of the ozone distribution."

In general, the mean ozone is biased high (see Table 2). Depending where one looks, the 20-year return level (roughly equivalent to a percentile of the ozone distribution) (see Table 3) is biased somewhat high but nevertheless, the width of the high end of the ozone distribution is underestimated. This is discussed in more detail in reference to Table 2.

Page 7, line 24: the 2nd "while" isn't needed

Fixed. Thank you.

Page 10, line 22: This sentence seems to be missing something: : : The greatest simulated differences?

Thank you. The sentence has been modified as follows:
Simulated differences between 20-year return temperatures and mean temperature (Figure \ref{fig:5retlvl}; Figure S4) are largest in the Northern part of the domain and extend southwards through the Midwest consistent with Figure \ref{fig:2temp} and Figure S1.

Page 11, line 20: Could EPA AQS data be added to fill some of these gaps?

We have also calculated the various metrics using the EPA AQS data in the revised manuscript. These results are included in the supplement. Unfortunately, we could only find approximately 120 measurement sites with both nearly complete temperature and ozone data for the period of interest (1992-2010).

Page 10, lines 10-20: A discussion on comparing the performance of REFC1SD to GCM2000 would be useful here and elsewhere, where appropriate – namely, the effect of having nudged meteorology rather than a free running simulation that may not simulate the synoptic conditions conducive to ozone and/or temperature extremes.

We prefer to postpone a general discussion of their differences to the discussion section, adding the following paragraph:

"Some of the differences between the CESM1-REFC1SD and GCM2000 simulations are likely due to meteorological differences:  while the CESM1-REFC1SD simulation is driven with analyzed meteorology, the GCM2000 simulation is driven with model calculated meteorology. In general, the CESM1-REFC1SD simulation captures the measured relation between ozone and

temperature better than the GCM2000 simulation over the Northeast US, although it does not fully capture their strong measured correlation (e.g., see Figure 6). In the Southeast the measured response appears to be generally well simulated in both simulations. Overall the GCM2000 fails to capture the extreme dependence as measured by PHI between temperature and ozone at the 95% level; in the CESM-REFC1SD simulation we cannot reject the null hypothesis that the simulated and measured values are the same."

In addition, in the discussion section we also discuss different simulations of the Bermuda high in the two simulations in the following paragraph.

Page 15, line 4: How are these sites chosen?

We've included the following in the paper:

These sites are selected to show a range of behavior in measured and simulated $\varphi$: in one site measured $\varphi$ is larger than simulated (Ashland, Me), less than that simulated (Sand Mountain, Al) or about the same as that simulated (Beaufort, NC).

These sites are selected to show a range of behavior in measured and simulated $\varphi$: at one site measured $\varphi$ is larger than simulated (Ashland, Me), at one site it is less than that simulated (Sand Mountain, Al), and at one site the measured and simulated values are about the same (Beaufort, NC)

Referee #3

Major Comments:
1. The authors provide an elegant methodology to minimize the influence of year-to-year variability and seasonal effects. At the same time this procedure comes at the cost of not considering all/many extremes on record. I am wondering about the effect the omission number a (here a=10) has on the results. How robust are the presented findings for values of a=5, a=15, a=1? Also I am curious if an approach omitting a certain number of extremes is more robust than more standard de-clustering approaches? It would be great if the authors could include a statement on this (or maybe an additional appendix) in the revised manuscript.

We state in the revised manuscript:
"We picked the number 10 originally as we wanted to preserve the extremes at approximately 10% of the points. As we have 92 days of data each summer (for JJA) we excluded 10 points from the analysis. As a sensitivity test we have omitted a different number of extreme points (5, 10 or 15) at a number of measurement sites to determine the difference in average ozone for each summer. These differences are on the order of 1 ppb, small compared to the data average."
(see figure below).

We did not try a de-clustering approach. The approach does not seem very sensitive to the number of points retained.

2. CAM4-chem is only one of the models contributing to the CCMI effort. In this light it would be great to see how other models that performed the REFC1SD experiment agree with observations. That said, I am not asking the authors to provide an in depth analysis across various models. It would be though interesting and provide important context for the community if the bias found for the model considered is 'common' in magnitude and sign across models and coherent across regions.

In the revised paper we include analysis from the other models contributing to CCMI in a supplement, but refer to this analysis where appropriate. Unfortunately, there are only a small number of additional models whose output includes the necessary fields to calculate the daily maximum temperature and the MDA8 ozone.

3. The authors use the set of CASTNET sites for comparison with model data, however we do not learn about which site selection criteria have been applied.

We have included the following in the revised manuscript:

"Hourly measured ozone and temperature data are taken from 23 CASTNET (Clean Air Status and Trends Network) stations with a nearly continuous data record during the period from 1992-2013 for the months of June, July, and August (92 days each summer). In addition, to enhance the data record, we included two additional stations (Beufort NC, and Lassen Volcanic CA) where the first 2 years or 3 years of data were missing, respectively. See Figure 2 for station locations. CASTNET sites are situated to sample regional ozone concentrations so as to minimize the more local impact of urban areas."

4. CASTNET data seems a robust choice to evaluate a global model for the US domain. However given the complications in the spatial correlation patterns of temperature and ozone I wonder if supplementing the observational data set with a suite of selected AQS sites might help. In the current analysis only 5 sites are used to cover the entire West, which raises robustness concerns.

We have also calculated the various metrics the EPA AQS data in the revised manuscript. These results are included in the supplement. Unfortunately, we could only find approximately 120 measurement sites with both nearly complete temperature and ozone data for the period of interest (1992-2010).

5. The authors choose three sites (Ashland, Sand Mountain, Beaufort) to illustrate ozone temperature correlations. How have those stations been chosen from the CASTNET set, and would it be not more intriguing to pool stations for the spatial domains indicated in Fig. 2a?

We have changed the text to refer to our reasons for picking these three stations:

These sites are selected to show a range of behavior in measured and simulated $\varphi$: in one site measured $\varphi$ is larger than simulated (Ashland, Me), less than that simulated (Sand Mountain, Al) or about the same as that simulated (Beaufort, NC).

These sites are selected to show a range of behavior in measured and simulated $\varphi$: at one site measured $\varphi$ is larger than simulated (Ashland, Me), at one site it is less than that simulated (Sand Mountain, Al), and at one site the measured and simulated values are about the same (Beaufort, NC)

6. The correlations reported are relatively low, which is acknowledged by the authors. I am wondering though if it would not be worthwhile to report explained variance throughout the manuscript instead of correlation coefficients.

We prefer to just stick with the correlations.

The key conclusions of the manuscript should be highlighted and also included in the abstract. Right now the language is a little vague regarding the significance and robustness of the overall extremal dependence. Despite all caveats raised in the discussion section, the study suggests a weak but robust relationship between ozone and temperature extremes. At the same time the spatial mismatch between regions where high ozone and high temperature extremes occur is of relevance and will motivate future modelling work

We have rewritten the abstract and a the conclusion at some length to help clarify the relationship between the temperature and ozone extremes. Hopefully these changes address the referees comments.

We have rewritten the last part of the abstract to read:

"Measures of spectral density are everywhere less than about 0.3, suggesting that at most only about a third of the time do very high temperatures coincide with very high ozone. Two regions of the U.S. have the strongest measured extreme dependence of ozone and temperature: the Northeast and the Southeast. The simulated future increase in temperature and ozone is primarily due to a shift in their distributions, not to an increase in their extremes. The locations where the right-hand side of the temperature distribution does increase (by up to 30%) are consistent with locations where soil-moisture feedback may be expected. Regional changes in the right-hand side of the ozone distribution range between +20% and -10%. The location of future increases in the high end tail of the ozone distribution are weakly related to those of temperature with a correlation of 0.3. However, the regions where the temperature extremes increase are not located where the extremes in ozone are large, suggesting a muted ozone response."

We also discuss the relationship between regions with large ozone and temperature extremes more thoroughly in the conclusion:

"All things being equal, we might expect the ozone distribution to be a slave to the temperature distribution so that regions of particularly high temperature extremes would be expected to have particularly high ozone extremes. However, this is not the case. High temperature extremes tend to occur in the Midwest while the highest ozone extremes tend to occur in the eastern part of the country. Regions with high ozone precursor emissions are known to increase the ozone-temperature slope so that regions with high precursor ozone emissions are particularly sensitive to temperature variations. Other complicating factors such as the importance of biogenic emissions or regional meteorological differences may also complicate the distributional relation between ozone and temperature."

We also clarify what a correlation of 0.3 implies between the increases in the right-hand side of temperature and ozone in the conclusions:

"In the future climate the ozone and temperatures distributions shift to the right. Our results generally suggest that the increase in both future temperature and future ozone is primarily due to a shift in the distributions, not to an increase in the extremes. Overall, the rhs of the temperature distribution increases slightly, with the largest increase, when evaluated at the CASTNET sites, in the Midwest. In some locations the increase in the rhs of the temperature distribution (mean(T|T>90%) - mean(T))approaches 30%, in other locations it decreases up to 10%. The pattern of increase is what might be expected from soil-moisture feedbacks. On average the width of the rhs of the future ozone distribution increases slightly by 0.26 ppb in the future simulation (mean(O3|O3>90%) - mean(O_3)), with regional increases up to 20% and decreases up to 10% over parts of the northeastern U.S. and much of the western two thirds of the country. When evaluated at CASTNET locations increases in the 20-year return period minus the mean are generally confined to the Midwest. The correlation between relative changes in the high end of the future temperature distribution (PSI(T,T)) and the ozone distribution (PSI(O3,O3)) is 0.3, relatively small but still significant. Thus an increase in the rhs of the future temperature distribution may have some impact on the future ozone distribution. However, the correlation is weak suggesting other complicating factors. It is possible that a stronger relationship would emerge from longer model simulations. In any case the region where the rhs

of the temperature distribution increases does not correspond to the region where the ozone extremes are relatively high. Elsewhere in the world, perhaps in regions with strong soil-moisture feedbacks and high emissions of ozone precursor emissions, a future amplification in the future temperature distribution would have more dramatic impacts on future ozone extremes."

Title: "Extremal dependence" might not be very accessible for non-statisticians.

Thank you. I think we will leave the title as is unless we come up with something better in the near future.

[Figure]

[Figure]

[Figure]

[Figure]

*Figure 1:* Difference in the average daily ozone deviation (ppb) depending on the number of points excluded from calculating the average JJA ozone concentration at A) Ashland Maine, B) Penn St, Pa. and C) Candor NC as a function of summer day (JJA) for 20 years. Blue shows the difference in omitting the 5 highest points versus omitting the highest point; red is the difference between omitting the 10 highest points versus the 5 highest points and green the difference between omitting the 15 highest points minus the 10 highest points. In all cases these differences are much less than the overall mean of the data